# Follow the MEP: Scalable Neural Representations for Minimum-Energy Path Discovery in Molecular Systems

## Abstract

Characterizing conformational transitions in physical systems remains a fundamental challenge, as traditional sampling methods struggle with the high-dimensional nature of molecular systems and high-energy barriers between stable states. These rare events often represent the most biologically significant processes, yet may require months of continuous simulation to observe. One way to understand the function and mechanics of such systems is through the minimum energy path (MEP), which represents the most probable transition pathway between stable states in the high-friction, low-temperature limit. We present a method that reformulates MEP discovery as a fast and scalable neural optimization problem. By representing paths as implicit neural representations and training with differentiable molecular force fields, our method discovers transition pathways without expensive sampling. Our approach scales to large biomolecular systems through a simple loss function derived from the path's likelihood via the Onsager-Machlup action and a scalable new architecture, AdaPath. We demonstrate this approach using four proteins, including an explicitly hydrated BPTI system with over 3,500 atoms. Our method identifies a MEP that captures the same conformational change observed in a millisecond-scale molecular dynamics (MD) simulation, obtaining this pathway in minutes on a standard GPU, rather than the weeks required on a specialized cluster.

## 1 Introduction

Molecular dynamics (MD) simulations provide detailed atomistic insight into biomolecular processes (Dror et al., 2012), effectively acting as a computational microscope. For large systems, exhaustive sampling of the Boltzmann distribution is infeasible, and the practical value of MD lies in capturing *individual* transition events that reveal mechanistic pathways and generate testable hypotheses for experimental validation. However, such events are exceedingly rare: transition rates between metastable states are exponentially suppressed by barrier heights (Eyring, 1935), causing trajectories to spend nearly all computational effort revisiting known stable states. As a consequence, observing even a single biologically relevant transition may require months of continuous simulation, motivating the development of more efficient methods for identifying transition mechanisms.

To characterize a transition mechanism without requiring prohibitively long MD trajectories, one can instead study the minimum-energy path (MEP), which represents the most probable route between stable states in the high-friction, low-temperature limit. Formally, a MEP is a continuous curve $\varphi : [0, 1] \to \mathbb{R}^{3N}$ connecting $x_A$ and $x_B$ and satisfying the orthogonality condition that the force has no normal component (E & Vanden-Eijnden, 2010). MEPs provide a compact representation of the transition by identifying intermediates, locating transition states, and offering a scaffold for sampling the entire transition-path ensemble.

Classical approaches approximate the path through a discrete set of images optimized in Cartesian coordinates. Chain-of-states methods (Pratt, 1986; Elber & Karplus, 1987; Ulitsky & Elber, 1990), the Nudged Elastic Band (NEB) and its variants (Jónsson et al., 1998; Henkelman et al., 2000), and string methods (E et al., 2002; 2007; Ren et al., 2005; Maragliano et al., 2006) all combine energy descent with either spring forces or periodic reparameterization. Although successful for small systems,

these formulations face well-known challenges: image collapse, sensitive spacing hyperparameters (Lindgren et al., 2019), and difficulties in constructing stable initial guesses (Ovchinnikov et al., 2011). More fundamentally, optimizing a discrete chain directly in Cartesian configuration space yields a stiff landscape, which often traps classical methods in poor local minima and makes it difficult to exploit the advantages of modern neural gradient-based optimization.

Beyond classical MEP solvers, many approaches aim to characterize transition mechanisms by sampling reactive trajectories. Traditional enhanced-sampling methods—such as transition path sampling, umbrella sampling, metadynamics, and related MD-based techniques (Bolhuis et al., 2002; Dellago et al., 1998; Laio & Parrinello, 2002; Torrie & Valleau, 1977; Petersen et al., 2024; Jung et al., 2023; Lazzeri et al., 2023), can reconstruct detailed transition-path ensembles but often incur substantial computational cost and typically require either a preexisting transition trajectory or well-chosen collective variables.

In parallel, machine-learning-based methods have introduced new ways to model transition pathways or ensembles. Variational formulations using Doob's $h$-transform (Das & Limmer, 2019; Singh & Limmer, 2023; Du et al., 2024; Lee et al., 2025), stochastic-control-based steering (Yan et al., 2022; Holdijk et al., 2023; Seong et al., 2025), and diffusion-model-based trajectory generators (Petersen et al., 2023; Han et al., 2024; Jing et al., 2024) each provide distinct mechanisms for circumventing brute-force sampling. However, these approaches often rely on complex multi-term objectives, involving higher-order differential terms known to hamper optimization (Rathore et al., 2024), or computationally heavy pretrained generative models. Closer to the present work are neural methods that explicitly parameterize transition paths (Ramakrishnan et al., 2025; Raja et al., 2025). While these methods avoid ensemble sampling, they typically use architectures that scale poorly with system size or require difficult-to-optimize losses.

Drawing inspiration from neural implicit representations (Sitzmann et al., 2020; Mildenhall et al., 2020), we revisit MEP discovery from a continuous optimization perspective. Starting from the Onsager–Machlup functional (Onsager & Machlup, 1953; Freidlin & Wentzell, 1998), which quantifies the probability of a stochastic trajectory, we derive a simple, single-term energy-based loss that depends only on potential energies evaluated along a continuous path, removing the need for higher-order derivatives and enabling efficient gradient-based optimization. Because this objective seeks to *minimize the potential energy along the path*, a naïvely parameterized network may reduce it by remaining near a single low-energy basin and introducing a discontinuous jump between states—a "teleportation" artifact arising directly from the energy-only structure of the loss. Rather than impose explicit continuity constraints such as springs or reparameterization, we instead rely on a neural path parameterization whose inductive biases encourage smooth variation along the path and make such discontinuities difficult to represent.

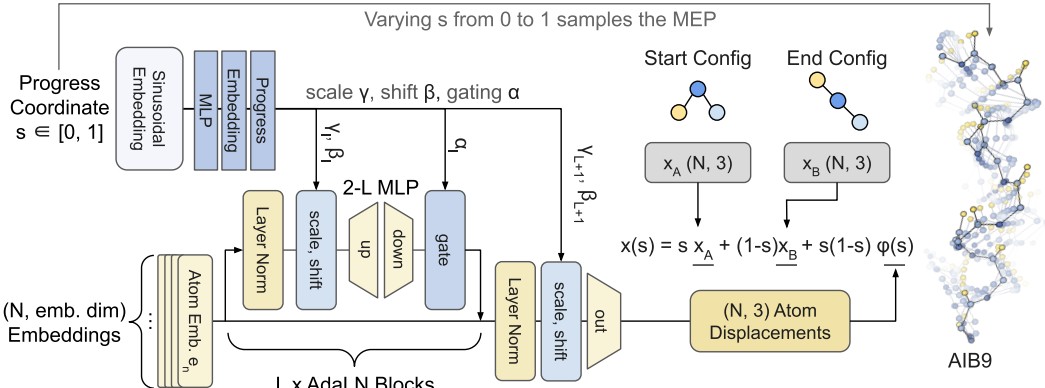

Figure 1: Overview of the AdaPath neural architecture and a generated MEP of the 129-atom AIB9 system. AdaPath transforms a progress coordinate $s \in [0, 1]$ into molecular configurations through learned atom embeddings processed by shared MLP blocks with adaptive conditioning. The conditioning network generates scale, shift, and gate parameters from $s$ to modulate each block. The final path $\phi(s) \in \mathbb{R}^{3N}$ continuously connects start and end states $x_A$ and $x_B$ via a simple interpolation corrected by the neural network.

To realize this in practice and ensure scalability, we introduce AdaPath 1, a neural architecture that represents the path using shared parameters across all atoms together with learned atom embeddings, combined with AdaLN-style residual MLP blocks (Peebles & Xie, 2023) that mix per-atom identity with global progress information. This design provides strong inductive biases toward coordinated, smooth deformations of the molecular configuration, suppressing discontinuous jumps and stabilizing optimization. These properties allow our formulation to extend naturally to explicitly solvated protein systems with thousands of atoms.

Using this formulation, our method discovers high-quality MEPs in minutes on a single GPU. We demonstrate this on systems ranging from alanine dipeptide and the AIB9 peptide to an explicitly hydrated BPTI system with over 3,500 atoms. In the BPTI case, the learned MEP reproduces the sequence of intermediates observed in a millisecond-scale MD simulation (Shaw et al., 2010), while requiring orders of magnitude fewer force-field evaluations.

## 2 THEORY AND METHODS

We first formalize the MEP problem and review established discrete methods. We then derive our continuous neural objective from the Onsager-Machlup action and introduce the AdaPath architecture designed to optimize this objective scalably.

### 2.1 CLASSICAL APPROACHES TO MEP FINDING

**Definition 1** (Minimum-Energy Path). *A curve $\varphi : [0, 1] \to \mathbb{R}^{3N}$ connecting $x_A$ and $x_B$ is a minimum-energy path (MEP) if $\left[\nabla U(\varphi(s))\right]^{\perp} = 0$ for all $s \in (0, 1)$, i.e., the force has no component orthogonal to the path tangent.*

MEP computation requires balancing two competing objectives: reducing the potential energy along the path while maintaining smooth, continuous transitions between the endpoints. Pure energy minimization generally produces discontinuous jumps between basins, whereas enforcing continuity alone forces the path through high-energy regions. Classical methods resolve this trade-off through either explicit continuity penalties or reparameterization-based constraints.

**Chain-of-states methods:** Chain-of-states formulations (Pratt, 1986; Elber & Karplus, 1987; Ulitsky & Elber, 1990) enforce continuity by penalizing differences between neighboring images. For a discretized path $\{x_0, \ldots, x_N\}$ with $x_0 = x_A$ and $x_N = x_B$, the update rule is

$$\dot{x}_i = -\nabla U(x_i) + k\left(x_{i+1} + x_{i-1} - 2x_i\right)/\Delta s, \tag{1}$$

where the spring term promotes uniform spacing. Large $k$ enforces continuity but induces "corner cutting"; small $k$ allows discontinuities.

The Nudged Elastic Band (NEB) method (Jónsson et al., 1998; Henkelman et al., 2000) resolves this by splitting the roles of physical and spring forces:

$$\dot{x}_i = -\left[\nabla U(x_i)\right]^{\perp} + k\left(x_{i+1} + x_{i-1} - 2x_i\right)^{\|}/\Delta s. \tag{2}$$

where $[\nabla U]^{\perp}$ represents forces perpendicular to the path and the spring term acts only parallel to it. NEB reduces corner cutting but may lead to jagged paths (Zhang et al., 2016).

**String methods:** String methods treat the path as a continuous curve $\varphi(s)$ and enforce the MEP condition directly (E et al., 2002; 2005; Ren et al., 2005; Maragliano et al., 2006). The idealized evolution is

$$\frac{\partial \varphi}{\partial t} = -\left[\nabla U(\varphi)\right]^{\perp}, \tag{3}$$

removing perpendicular force components while permitting free reparameterization along the tangent. Implementations discretize $\varphi$ into images and alternate: (i) gradient descent $\dot{x}_i = -\nabla U(x_i)$, and (ii) reparameterization to maintain equal arc length:

$$s_i = \frac{\sum_{j=0}^{i-1} \|x_{j+1} - x_j\|}{\sum_{j=0}^{N-1} \|x_{j+1} - x_j\|}. \tag{4}$$

**Computational limitations:** The above methods differ in how they enforce continuity, but all optimize discrete images directly in Cartesian space. This leads to stiffness, sensitivity to initialization and hyperparameters (Ovchinnikov et al., 2011; Lindgren et al., 2019), and limited opportunities for parameter sharing or parallelization. These constraints motivate continuous neural path representations and objective functions that avoid explicit spring terms or repeated reparameterization.

## 2.2 DERIVATION OF THE ENERGY-BASED LOSS FUNCTION

**From Langevin dynamics to the OM functional:** The limitations of classical discrete methods motivate our continuous neural formulation. To derive our loss function, which, when minimized, yields a continuous curve representing the MEP connecting fixed endpoints $x_A$ and $x_B$, we build upon alternative formulations of the Onsager-Machlup action functional as explored by Vanden-Eijnden & Heymann (2008) and Olender & Elber (1997).

To introduce the action intuitively, we begin with a molecular system evolving under overdamped Langevin dynamics, a model for molecular motion in the high-friction regime characteristic of biomolecular systems:

$$\dot{x}(t) = -\nabla U(x(t)) + \sqrt{2}\eta(t) \tag{5}$$

where $U(x)$ represents the potential energy function and $\eta(t)$ is Gaussian white noise modeling thermal fluctuations. To derive the path probability, we discretize the time interval $[0, T]$ into $N$ steps of duration $\Delta t$, approximating the dynamics as $x_{i+1} = x_i - \Delta t \nabla U(x_i) + \sqrt{2\Delta t}\eta_i$. The conditional probability of transitioning from $x_i$ to $x_{i+1}$ follows a Gaussian distribution:

$$P(x_{i+1}|x_i) \propto \exp\left(-\frac{(x_{i+1} - x_i + \Delta t \nabla U(x_i))^2}{4\Delta t}\right) \tag{6}$$

Taking the product of these conditional probabilities over all time steps $N$ and then taking the limit as $\Delta t \to 0$ yields the path probability $\mathbb{P}[x(t)] \propto \exp(-\frac{1}{4}\int_0^T \|\dot{x}(t) + \nabla U(x(t))\|^2 dt)$. The exponent corresponds to the Onsager-Machlup action functional.

**Definition 2** (Onsager-Machlup Action). *The action functional governing the probability of a trajectory $x(t)$ is defined as (Onsager & Machlup, 1953):*

$$S_{\text{OM}}[x] = \frac{1}{4}\int_0^T \|\dot{x}(t) + \nabla U(x(t))\|^2 dt \tag{7}$$

*Thus, maximizing the probability of a transition path (subject to fixed endpoints $x(0) = x_A$ and $x(T) = x_B$) is equivalent to minimizing this action functional.*

Notably, there are different derivations of the action that yield an additional higher-order derivative term related to trajectory entropy, whose necessity is conditional on the task (Adib, 2008). Since we are primarily interested in MEPs, we neglect this term. However, the resulting MEPs still empirically align well with the free-energy landscape. We note that this might not hold for strongly entropy-driven systems, such as intrinsically disordered proteins (Granata et al., 2015; Caro et al., 2017; Skriver et al., 2023).

**Simplifying the action and removing the fixed time parameterization:** We expand the integrand of the action functional:

$$\|\dot{x}(t) + \nabla U(x(t))\|^2 = \|\dot{x}(t)\|^2 + 2\dot{x}(t) \cdot \nabla U(x(t)) + \|\nabla U(x(t))\|^2 \tag{8}$$

For paths with fixed endpoints, the cross-term integrates to a constant difference in potential energy $\int_0^T \dot{x}(t) \cdot \nabla U(x(t))dt = U(x_B) - U(x_A)$. Since $x_A$ and $x_B$ are fixed, this term does not affect the optimization. Applying the Cauchy-Schwarz inequality to the remaining terms:

$$\|\dot{x}(t)\|^2 + \|\nabla U(x(t))\|^2 \geq 2\|\dot{x}(t)\|\|\nabla U(x(t))\| \tag{9}$$

This upper bound becomes an equality when $\dot{x}(t)$ is parallel or antiparallel to $\nabla U(x(t))$, i.e., when the path follows the gradient. Crucially, we allow the path to optimize its shape and length, enabling it to align with energy gradients and making the upper bound tight at the optimum.

When we abandon fixed-time parameterization and allow the path to stretch as needed, we can transform the time integral into a purely geometric one by defining the arc-length element $ds_{arc} = \|\dot{x}(t)\| dt$ along the spatial curve $\varphi$.

**Proposition 1** (Geometric Action). *By relaxing the time parameterization, the minimization problem reduces to a geometric integral over the curve $\varphi$:*

$$\int_0^T \|\nabla U(x(t))\|\|\dot{x}(t)\|dt = \int_\varphi \|\nabla U(x)\|ds_{arc} \tag{10}$$

**Discretization and first-order approximation:**    While classical chain-of-states methods minimize this geometric integral via direct Cartesian updates, we seek a loss function that acts directly on the neural network parameters $\theta$. To avoid computing gradients along the path (which introduces optimization challenges), we discretize the path into segments $\Delta\varphi_k = \varphi_{k+1} - \varphi_k$ and simplify the geometric action $\int_\varphi \|\nabla U(x)\|ds_{arc} \approx \sum_{k=0}^{K-2} \|\nabla U(\varphi_k)\|\|\Delta\varphi_k\|$.

**Assumption 1** (First-Order Taylor Approximation). *We assume the displacement $\|\Delta\varphi_k\|$ is sufficiently small such that the potential energy difference is dominated by the first-order term:*

$$U(\varphi_{k+1}) - U(\varphi_k) = \nabla U(\varphi_k) \cdot \Delta\varphi_k + O(\|\Delta\varphi_k\|^2) \approx \|\nabla U(\varphi_k)\|\|\Delta\varphi_k\|\cos\theta_k \tag{11}$$

*where $\theta_k$ is the angle between the gradient and displacement vectors.*

For MEPs, the displacement aligns with the gradient direction ($\cos\theta_k \to \pm 1$). Provided the higher-order term is negligible, this yields $\|\nabla U(\varphi_k)\|\|\Delta\varphi_k\| \approx |U(\varphi_{k+1}) - U(\varphi_k)|$. This gives us a surrogate objective with the same minimum.

**Proposition 2** (Energy Upper Bound). *Using the triangle inequality on the absolute energy differences, we establish that minimizing the simple sum of energies along the path provides an upper bound on the action:*

$$\sum_{k=0}^{K-2} |U(\varphi_{k+1}) - U(\varphi_k)| \leq \sum_{k=0}^{K-1} U(\varphi_k) - (K-1)\min_k U(\varphi_k) - \frac{U(\varphi_0) + U(\varphi_{K-1})}{2} \tag{12}$$

*where $\min_k U(\varphi_k)$ is the lowest energy along the path. Since the latter terms are constants during optimization, this leads to our final loss function.*

**The Neural Objective:**    To implement this in practice, we sample $B$ points along a parametrized path $\varphi_\theta(s)$ and directly minimize the sum of energies at these points.

**Definition 3** (Energy-Based Loss Function). *We define the optimization objective $\mathcal{L}(\theta)$ as the expected potential energy approximated via Monte Carlo sampling:*

$$\mathcal{L}(\theta) = \frac{1}{B}\sum_{j=1}^{B} U(\varphi_\theta(s_j)), \quad s_j \sim \mathcal{U}[0,1] \tag{13}$$

**Remark 1** (Preventing Pathological Solutions). *While this energy-based loss successfully identifies MEPs when Assumption 1 holds, it admits a pathological minimum when it breaks down. Specifically, if $\|\Delta\varphi_k\|$ is large, the Taylor series approximation fails. In that case, the loss can be minimized by paths that remain in one stable state and then rapidly "teleport" to the other, avoiding the high-energy transition region. We employ specific architectural choices (AdaPath) and optimizer selection to control $\|\Delta\varphi_k\|$ and prevent this behavior.*

### 2.2.1 ADAPATH: A SCALABLE NEURAL ARCHITECTURE FOR MEPS

Standard neural parameterizations mapping $s \in [0,1]$ directly to $\mathbb{R}^{3N}$ require hidden layers that scale quadratically with system size ($O(N^2)$). AdaPath (Figure 1) circumvents this by disentangling the path progress information from the atomic representation, using a shared network conditioned by $s$.

**Learnable Atom Embeddings:**    We assign a learnable embedding vector $\mathbf{e}_n \in \mathbb{R}^d$ to each atom $n$. This creates a parameter set that scales linearly with $N$. One can interpret the learned vectors as encoding types of motion along the MEP.

$$\mathbf{H}^{(0)} = [\mathbf{e}_1, \ldots, \mathbf{e}_N]^\top \in \mathbb{R}^{N \times d} \tag{14}$$

**Progress Conditioning:**    An AdaLN-like (Peebles & Xie, 2023) conditioning network processes the sinusoidal embedding of $s$ to generate modulation parameters for the entire system. This MLP (width $d_c$) outputs a vector containing all scale ($\gamma_l$), shift ($\beta_l$), and gate ($\alpha_l$) parameters for all $L$ backbone layers and the final projection.

$$\mathbf{p}(s) = \text{MLP}_{\text{cond}}(\text{Sinusoidal Embedding}(s)) \in \mathbb{R}^{L \times 3d + 2d} \tag{15}$$

This network introduces a constant parameter overhead independent of $N$, amortizing the cost of generating complex path dynamics across all atoms.

**AdaLN Blocks:**    The shared backbone comprises $L$ identical blocks applied to each atom. Each block modulates the atom's hidden state $\mathbf{h}_n^{(l)}$ using the global conditioning parameters via Adaptive Layer Normalization (AdaLN), paired with a standard Transformer-style MLP block with latent dimensionality expansion factor $E$:

$$[\gamma_l, \beta_l, \alpha_l] = \text{Split}(\mathbf{p}_l(s)) \quad \text{(Sliced from } \mathbf{p}(s)) \tag{16}$$

$$\hat{\mathbf{h}} = \text{Norm}(\mathbf{h}_n^{(l)}) \odot (1 + \gamma_l) + \beta_l \tag{17}$$

$$\mathbf{h}_{\text{up}} = \text{GELU}(\mathbf{W}_{\text{up}}\hat{\mathbf{h}}), \quad \mathbf{W}_{\text{up}} \in \mathbb{R}^{Ed \times d} \tag{18}$$

$$\mathbf{h}_n^{(l+1)} = \mathbf{h}_n^{(l)} + \alpha_l \odot \mathbf{W}_{\text{down}}\mathbf{h}_{\text{up}}, \quad \mathbf{W}_{\text{down}} \in \mathbb{R}^{d \times Ed} \tag{19}$$

After $L$ blocks, a final normalization and specific scale/shift correction ($\gamma_{\text{final}}, \beta_{\text{final}}$) are applied before the projection to Cartesian coordinates:

$$\varphi_n(s) = \mathbf{W}_{\text{out}} \left( \text{Norm}(\mathbf{h}_n^{(L)}) \odot \gamma_{\text{final}} + \beta_{\text{final}} \right) \tag{20}$$

**Endpoint Constraints.**    To enforce boundary conditions $x(0) = x_A$ and $x(1) = x_B$, we employ a linear interpolation between endpoints with a quadratic neural correction, which vanishes at the end points:

$$x(s) = (1 - s)x_A + sx_B + s(1 - s)\varphi_\theta(s) \tag{21}$$

**Parameter Efficiency:**    AdaPath radically reduces the memory footprint for large systems compared to standard MLPs. The total parameter count is:

$$N_{\text{params}} \approx \underbrace{N \cdot d}_{\text{Atom Embeds}} + \underbrace{L \cdot 2Ed^2}_{\text{Shared Backbone}} + \underbrace{O(d_c^2)}_{\text{Conditioning Net}} \tag{22}$$

Crucially, the $O(N^2)$ scaling term is eliminated. For BPTI ($N \approx 3500$, $d = 64$, $E = 2$), a standard MLP, with uniform hidden dimesnions, would require $\sim$1.4 billion parameters, whereas AdaPath requires only $\sim$3.5 million, with the majority of parameters residing in the fixed-size conditioning network rather than scaling with the protein size.

**Continuity and Optimization:**    The residual MLP design enables gradient flow and maintains path continuity through the architectural inductive bias of smooth function approximation. While our energy-based loss does not explicitly enforce continuity, the neural network's implicit smoothness prevents discontinuous solutions. However, standard optimizers can cause excessive path stretching around transition regions (Figure 8). We therefore use the Muon optimizer (Keller Jordan et al., 2024) for small systems, as it maintains lower Lipschitz constants (Large et al., 2024), ensuring adequate path presence in transition regions. For large systems, Adam variants (Kingma & Ba, 2017; Dozat, 2016) suffice; we posit that neural networks' increasing inability to model discontinuities as dimensionality increases (Petersen & Voigtlaender, 2018) naturally enforces continuity. **We empirically validate this behavior in Appendix B.7**, demonstrating that our architectural and optimization choices maintain path velocity variations comparable to classical string methods (Table 10).

## 3   EXPERIMENTS

We evaluate AdaPath across a hierarchy of molecular systems to assess both benchmark precision and high-dimensional scalability. We compare our method against a suite of classical MEP approaches

(Chain-of-States, String Method, NEB) and recent neural baselines (Doob's Lagrangian (Du et al., 2024), Doob's Seq2Seq (Lee et al., 2025)).

**Roadmap to Appendices:** To ensure reproducibility, we provide comprehensive experimental details in the Appendices. Detailed benchmarks and hyperparameter sweeps are provided for alanine dipeptide (Appendix B.3), AIB9 (Appendix B.4), and BPTI (Appendix B.5). To validate path continuity and optimization speed, we provide path velocity analysis (Appendix B.7) and wall-clock timing benchmarks (Appendix B.8). Component contributions are analyzed in ablation studies (Appendix D): architectural validation confirms parameter sharing is essential for scalability, optimizer analysis reveals Muon maintains stable path continuity, and loss function comparisons validate our energy-based objective against INR formulations. Finally, we validate the physical relevance of our pathways via committor analysis (Appendix G.1) and provide additional validation on the Villin HP35 system in Appendix G.

### 3.1 BENCHMARK COMPARISON ON ALANINE DIPEPTIDE

To provide a rigorous comparison with existing methods, we benchmark AdaPath on the standard alanine dipeptide system using experimental conditions matching Du et al. (2024) and reference their results: AMBER14 force field at 300 K. To ensure a comprehensive evaluation, we include a wide range of baselines: MCMC sampling, classical MEP approaches (Chain-of-States, String Method, NEB), and recent neural methods including Doob's Lagrangian variants (Du et al., 2024), MaxLL/Seq2Seq (Lee et al., 2025), and Implicit Neural Representations (INR) (Ramakrishnan et al., 2025).

Table 1: Comprehensive benchmark on alanine dipeptide. We compare against MCMC baselines, Doob's Lagrangian variants (Du et al., 2024), MaxLL/Seq2Seq (Lee et al., 2025), and the INR method (Ramakrishnan et al., 2025). To isolate the impact of the loss function, we include **AdaPath w/ INR Loss**. **Time** denotes the wall-clock time required to reach the reported best energy. **Max Energy** is the mean $\pm$ std transition barrier. **MinMax** is the best barrier found across runs. **Mean Energy** is arc-length adjusted. Energies in kJ/mol. Best results are **bolded**.

| Method | # Evals | Time (m) | Max Energy | MinMax | Mean Energy |
|---|---|---|---|---|---|
| *MCMC Baselines* | | | | | |
| MCMC var. len. | 2.1e7 | – | $740 \pm 700$ | 52.4 | N/A |
| MCMC* | 1.3e9 | – | $288 \pm 128$ | 60.5 | N/A |
| *Doob's Methods* | | | | | |
| Doob's Cartesian | 3.8e7 | – | $726.40 \pm 0.07$ | 726.2 | N/A |
| w/ 5 Mixtures | 5.1e7 | – | $541 \pm 278$ | 248.0 | N/A |
| Doob's Internal | 3.8e7 | – | $-14.62 \pm 0.02$ | $-14.67$ | N/A |
| w/ 5 Mixtures | 5.1e7 | – | $-15.5 \pm 0.3$ | $-15.95$ | N/A |
| Doob's Seq2Seq | – | – | $3.46 \pm 0.03$ | 3.46 | N/A |
| *Single-Path MEP Methods (Classical & Neural)* | | | | | |
| AdaPath w/ INR Loss | 1.1e6 | 3.30 | $\mathbf{-80.54 \pm 0.33}$ | $-81.02$ | $-83.9 \pm 0.8$ |
| Chain-of-States | 2.4e5 | **0.03** | $-71.1$ | $-71.1$ | $-82.9$ |
| String Method | 3.3e6 | 0.44 | $-67.7$ | $-67.7$ | $\mathbf{-85.3}$ |
| NEB | 7.7e6 | 0.82 | $-67.7$ | $-67.7$ | $-83.5$ |
| **AdaPath (ours)** | **6.4e4** | 0.24 | $-80.26 \pm 0.20$ | $\mathbf{-86.58}$ | $-85.1 \pm 1.6$ |

**Impact of Loss Function:** To disentangle the benefits of our architecture from our optimization objective, we explicitly benchmark "AdaPath w/ INR Loss." This configuration uses our scalable architecture but is trained using a NEB-inspired loss function proposed by Ramakrishnan et al. (2025); full details are provided in Appendix D.5. As shown in Table 1, while the INR loss yields a competitive mean barrier, our simple energy-based loss allows AdaPath to discover significantly deeper minima (MinMax Energy of $-86.6$ vs. $-81.0$ kJ/mol). Furthermore, the INR loss optimization is significantly more expensive, requiring 3.3 minutes to reach these results compared to 0.24 minutes for our energy-based loss.

**Efficiency vs. Quality:** Table 1 reports both total force evaluations and wall-clock time. Crucially, the reported time represents the duration required to reach the specific lowest energy reported for that method. Consequently, a very short wall-clock time may indicate premature convergence to a suboptimal solution rather than algorithmic superiority. For instance, Chain-of-States is extremely fast (0.03 min) due to simple force-based Cartesian updates, but it converges to a poor pathway (Max Energy $-71.1$ kJ/mol). In contrast, AdaPath requires more time (0.24 min) due to neural network backpropagation but discovers substantially better transition mechanisms (Max Energy $-86.6$ kJ/mol).

## 3.2 Scalability to High-Dimensional Systems

Having validated the method on the standard benchmark, we now examine AdaPath's scalability to higher-dimensional systems where classical initialization often fails. We focus on the AIB9 peptide and the BPTI protein.

### 3.2.1 AIB9 Transition Path Discovery

We first examine the AIB9 system, a 9-residue artificial protein with 129 atoms. It exhibits two well-defined metastable states, making it an ideal test case for transition path methods: it is small enough for extensive reference simulations yet complex enough to exhibit realistic conformational changes. We use the AMBER ff15ipq-m force field for protein-mimetics (Bogetti et al., 2020) implemented in DMFF (Wang et al., 2023).

Table 2: Benchmark results on the AIB9 peptide. **Time** denotes wall-clock time to convergence. AdaPath discovers significantly deeper energy minima ($-644$ vs. $-461$ kJ/mol) than the best classical method. While classical methods are faster per step, they converge to suboptimal local minima or fail to find the transition entirely. Doob's Lagrangian results are reproduced from Du et al. (2024).

| Method | # Evals | Time (m) | Max Energy | MinMax | Mean Energy |
|---|---|---|---|---|---|
| Chain-of-States | 2.9e6 | **0.9** | $-461.0$ | $-461.0$ | $-687.2$ |
| String Method | 3.1e6 | 0.7 | 1.3e4 | 1.3e4 | 3.2e3 |
| NEB | 4.1e6 | 1.0 | 2.7e3 | 2.7e3 | $-128.0$ |
| Doob's Lagrangian | 1.2e7 | – | $1.6e4 \pm 7.8e3$ | 5.8e3 | N/A |
| **AdaPath (ours)** | 4.7e6 | 18.9 | $\mathbf{-496 \pm 179}$ | $\mathbf{-644.0}$ | $\mathbf{-767 \pm 24}$ |

Figure 2 shows the discovered transition paths projected onto the central residue's $\phi$-$\psi$ dihedral angle space. The method identifies multiple physically plausible pathways between the states. Table 2 quantifies the performance: AdaPath achieves a MinMax barrier of $-644$ kJ/mol, substantially outperforming Chain-of-States ($-461$ kJ/mol) and other baselines which effectively fail to find the low-energy channel (positive barriers $> 10^3$ kJ/mol). While AdaPath requires more wall-clock time (18.9 min) than the fast-but-suboptimal classical runs ($\sim 1$ min), it is the only method that reliably identifies the true low-energy transition mechanism, whereas baselines rapidly converge to poor local minima.

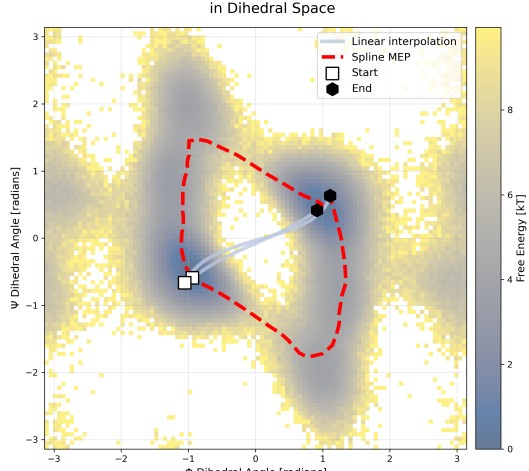

Figure 2: AIB9 free energy surface projected onto central residue dihedral angles ($\phi, \psi$). Two distinct MEPs were found for different start/end points and model initializations, demonstrating the method's ability to find multiple distinct transition pathways.

### 3.2.2 BPTI Conformational Change Pathway

To test the limits of scalability, we applied AdaPath to Bovine Pancreatic Trypsin Inhibitor

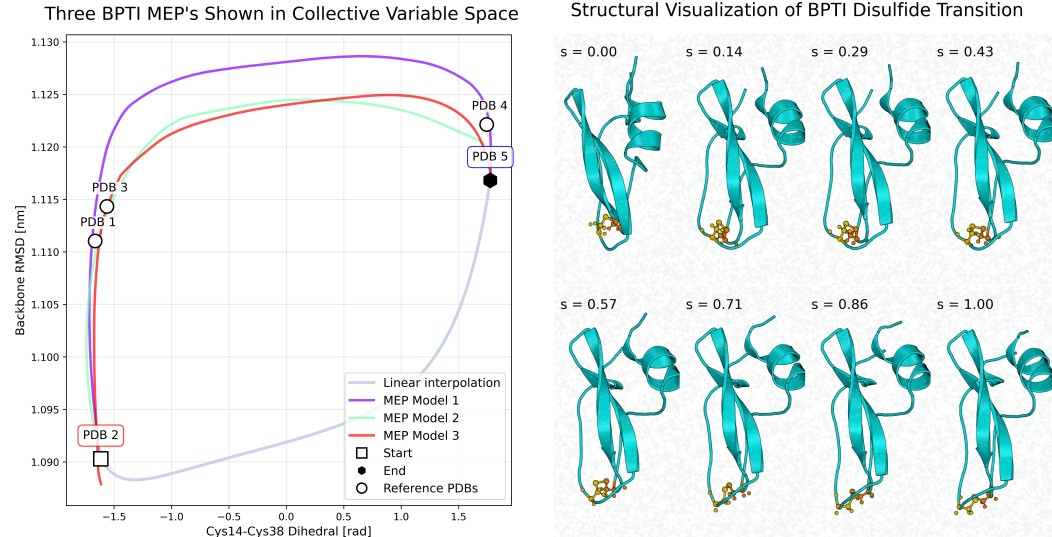

Figure 3: Analysis of our discovered BPTI transition pathway. **Left:** Evolution of BPTI dynamics projected onto two collective variables: the Cys14-Cys38 disulfide torsion angle versus the mean-centered backbone RMSD. Reference structures from D.E. Shaw simulation are marked with circles. **Right:** Structural visualization of the BPTI and water system highlighting the Cys14-Cys38 disulfide bond conformation along one of our optimized transition paths.

(BPTI). BPTI is 58 residues long (892 atoms) and exhibits complex conformational changes involving disulfide bond rearrangements. We match the experimental conditions of a landmark millisecond-scale MD simulation by D.E. Shaw Research (Shaw et al., 2010), incorporating explicit water molecules (raising the system size to over 3,500 atoms).

Table 3: Benchmark results on explicitly hydrated BPTI (3,500+ atoms). AdaPath outperforms classical methods by an order of magnitude in both energy minimization and sample efficiency. Notably, AdaPath finds a valid transition path in $\sim 9$ minutes with $< 10^5$ evaluations, whereas classical methods struggle to converge or produce high-energy jagged paths despite $10\times$ more evaluations.

| Method | # Evals | Time (m) | Max Energy | MinMax | Mean Energy |
|---|---|---|---|---|---|
| Chain-of-States | 1.4e6 | 10.8 | $-1.3$e4 | $-1.3$e4 | $-2.0$e4 |
| String Method | 3.8e6 | 28.0 | 1.4e4 | 1.4e4 | 4.3e3 |
| NEB | 1.1e6 | **8.2** | 1.6e5 | 1.6e5 | 2.3e5 |
| **AdaPath (ours)** | **9.6e4** | 9.1 | $\mathbf{-7.1e4 \pm 4.1e4}$ | $\mathbf{-1.4e5}$ | $\mathbf{-2.2e5 \pm 1.9e5}$ |

Remarkably, as shown in Figure 3, our optimized path passes through all intermediate snapshots, indicating that it successfully discovers the same transition mechanism as the computationally intensive MD simulation. Table 3 highlights the massive efficiency gap on this high-dimensional system. AdaPath discovers a path with a maximum energy of $-1.4$e5 kJ/mol using only $9.6 \times 10^4$ evaluations. In contrast, classical methods require over $10^6$ evaluations and either fail to converge (String Method, $+1.4$e4 kJ/mol) or produce extremely high-energy jagged paths (NEB, $+1.6$e5 kJ/mol). In terms of wall-clock time, AdaPath is highly competitive (9.1 min), matching the speed of NEB while delivering vastly superior path quality. This represents a speedup of several orders of magnitude compared to the weeks of specialized cluster time required for the original equilibrium MD simulation.

### 3.3 ABLATION STUDIES AND ANALYSIS

To isolate the contributions of specific algorithmic components to the efficiency reported above, we conducted extensive ablation studies (fully detailed in Appendix D).

**Architectural Validation:** Parameter sharing proves essential for scalability; standard MLPs fail catastrophically on large systems, with even billion-parameter variants showing poor performance on BPTI. **Optimization and Continuity:** Our optimizer analysis reveals that Muon maintains stable path continuity, while Adam exhibits erratic "teleportation" behavior that rapidly jumps between states without traversing transition regions in small systems. Quantitative analysis of this behavior via path velocity metrics is provided in Appendix B.7, and wall-clock performance comparisons are provided in Appendix B.8. **Loss Function:** The logarithmic energy transformation proves critical for training stability, reducing energy variance by orders of magnitude. Comparison with INR loss formulations (Ramakrishnan et al., 2025) (see Appendix D) confirms that our simple energy-based objective yields better results for large systems.

## 4 LIMITATIONS

While our approach demonstrates strong performance on protein systems up to 3,500 atoms, several limitations should be noted. First, our method is constrained to systems with suitable differentiable force fields, though this limitation is diminishing as more implementations become available. Second, our method produces single representative MEPs rather than sampling the complete statistical ensemble of transition pathways. While this represents a trade-off between computational efficiency and exhaustive sampling, the mechanistic insights provided are often the primary objective for practitioners studying biomolecular transitions.

## 5 CONCLUSION

We demonstrated that reformulating MEP discovery as a continuous neural optimization problem, combined with a scalable architecture, enables the efficient discovery of molecular transition mechanisms for large, explicitly solvated systems. Our benchmarks reveal a critical performance gap in existing approaches: while classical methods struggle to converge on high-dimensional landscapes and prior neural methods face scalability bottlenecks, our approach consistently identifies lower-energy barriers with orders of magnitude fewer force evaluations. Crucially, this efficiency translates into a qualitative shift in accessibility: by recovering the BPTI transition mechanism, a process spanning milliseconds, in minutes on a standard GPU, we show that biologically significant rare events can now be characterized without the prohibitive cost of specialized supercomputing.

Our method makes two key contributions: deriving a simple energy-based loss function directly from the Onsager-Machlup action functional, and introducing AdaPath. This architecture scales favorably with system size via parameter sharing across the atom dimension. By encoding path continuity through architectural design rather than explicit loss terms, we sidestep known optimization challenges in PINNs (Raissi et al., 2019; Rathore et al., 2024) and multi-loss training.

Several promising directions for future development emerge from this work. The scaling properties of AdaPath suggest the method could be extended to even larger biomolecular systems, such as membrane proteins or multi-domain complexes that routinely contain hundreds of thousands of atoms in MD simulations. Furthermore, our discovered MEPs could serve as guidance paths for enhanced sampling methods, enabling free energy calculations through escorted Jarzynski-like estimators (Jarzynski, 1997; Vaikuntanathan & Jarzynski, 2011) by using the MEP as a basis for a non-equilibrium pulling protocol between stable states. The architecture and loss function may also prove valuable for other non-biological physical systems where rare events might be of interest.

## 6 ETHICS STATEMENT

Our research democratizes access to molecular transition pathway studies by enabling complex biomolecular simulations on standard hardware, potentially accelerating drug discovery while contributing to sustainable computing through dramatically reduced energy consumption. We anticipate primarily positive societal impacts but acknowledge that accelerated protein engineering capabilities could have dual-use implications and encourage responsible application with appropriate safety considerations.

## 7 REPRODUCIBILITY STATEMENT

To ensure reproducibility of our results, we provide comprehensive implementation details throughout the paper and appendices. Section 2.2 contains the complete mathematical derivation of our energy-based loss function, with the extended mathematical treatment provided in Appendix A.2. The AdaPath architecture is fully specified in Section 2.1.1 with all hyperparameters detailed in the experimental sections. Detailed benchmark protocols are provided in Appendix B.3 for alanine dipeptide, Appendix B.4 for AIB9 peptide, and Appendix B.5 for BPTI systems, including complete hyperparameter optimization procedures for all classical methods. For the AIB9 system, our Doob's Lagrangian benchmark results can be reproduced using the publicly available implementation with the specific parameter settings detailed in Appendix B.4. Appendix D provides extensive ablation studies with complete parameter settings, with general experimental configurations specified in Appendix D.1. System preparation protocols are comprehensively documented in Appendix F for BPTI experiments, including force field specifications, explicit solvation procedures, and training schedules. We provide the complete source code for all experiments, including the AdaPath implementation, training procedures for AIB9 and BPTI systems, benchmark implementations for all classical methods, ablation study code, and hyperparameter sweep implementations. All molecular systems use standard, publicly available force fields (AMBER ff15ipq-m, AMBER99sb-ILDN, TIP3P) and reference structures from published datasets, with complete simulation protocols provided in the respective appendix sections.

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

# A APPENDIX

## A.1 LARGE LANGUAGE MODEL USAGE

Large Language Models (LLMs) were used as assistive tools during the preparation of this manuscript in the following limited capacities:

- **Spelling and grammar checking**: LLMs were used to identify and correct typographical errors, grammatical mistakes, and improve sentence structure throughout the manuscript.
- **Writing style suggestions**: LLMs provided recommendations for improving clarity, conciseness, and readability of technical explanations and descriptions.
- **Reformatting assistance**: LLMs helped with standardizing formatting, citation styles, and ensuring consistency in presentation across sections.
- **Coding assistance**: LLMs integrated within code editors provided suggestions for code completion, syntax correction, and implementation of standard programming patterns.

We take full responsibility for all content in this manuscript, including any LLM-assisted revisions.

## A.2 EXTENDED DERIVATION OF THE ENERGY-BASED LOSS FUNCTION

In this appendix, we provide a comprehensive derivation of our energy-based loss function for discovering minimum-energy paths (MEPs) between molecular configurations. The derivation proceeds in four main steps: (1) establishing the path probability density from overdamped Langevin dynamics, (2) reformulating this probability in terms of the Onsager-Machlup action functional, (3) simplifying this functional to a geometric line integral, and (4) developing a discretized approximation suitable for neural network optimization.

Our goal is to find the most probable transition path between two stable molecular configurations $x_A$ and $x_B$. This path corresponds to the minimum-energy pathway through the potential energy landscape and represents the most likely mechanism of conformational change.

### ASSUMPTIONS AND APPROXIMATIONS

The derivation relies on several key assumptions:

- The molecular system evolves according to overdamped Langevin dynamics, appropriate for high-friction biomolecular environments
- The path endpoints $x_A$ and $x_B$ are fixed and correspond to stable states
- The potential energy function $U(x)$ is differentiable everywhere along the path
- We can represent the path as a continuous, parameterized curve in configuration space with sufficient flexibility
- The path's parameterization allows for effective length changes, permitting the path to adjust its stretching as needed to follow energy gradients optimally

We also make several important approximations during the derivation:

- We use a first-order Taylor expansion to relate energy differences to gradient magnitudes, which requires displacement vectors $\|\Delta\varphi_k\|$ to be sufficiently small
- We introduce stochastic sampling to estimate the continuous energy integral
- For a MEP, the displacement tends to align either parallel or antiparallel with the gradient direction (i.e., $\cos\theta_k \to \pm 1$ for path segments)
- We use the triangle inequality to establish an upper bound on the sum of absolute energy differences

While the last two approximations do not change the location of the global minimum, they do affect the optimization landscape before that minimum is reached. Critically, when the Taylor

approximation breaks down due to large displacement vectors, the resulting energy-based loss can admit pathological solutions where paths remain in stable states and then rapidly "teleport" to avoid high-energy transition regions.

### A.2.1 FROM THE ONSAGER-MACHLUP ACTION TO A GEOMETRIC LINE INTEGRAL

We begin with a molecular system evolving under overdamped Langevin dynamics, a standard model for high-friction molecular motion:

$$\dot{x}(t) = -\nabla U(x(t)) + \sqrt{2}\eta(t) \tag{23}$$

where $x(t) \in \mathbb{R}^{3N}$ represents the molecular configuration, $U(x)$ is the potential energy function, and $\eta(t)$ is a Gaussian white noise with $\langle \eta_i(t)\eta_j(t') \rangle = \delta_{ij}\delta(t - t')$.

To derive the path probability, we discretize the time interval $[0, T]$ into $N$ steps of duration $\Delta t$, approximating the dynamics as:

$$x_{i+1} = x_i - \Delta t \nabla U(x_i) + \sqrt{2\Delta t}\eta_i \tag{24}$$

where $\eta_i$ are independent standard Gaussian random variables with zero mean and unit variance.

The conditional probability of transitioning from $x_i$ to $x_{i+1}$ follows a Gaussian distribution:

$$P(x_{i+1}|x_i) = \frac{1}{\sqrt{4\pi\Delta t}^d} \exp\left(-\frac{(x_{i+1} - x_i + \Delta t \nabla U(x_i))^2}{4\Delta t}\right) \tag{25}$$

$$\propto \exp\left(-\frac{(x_{i+1} - x_i + \Delta t \nabla U(x_i))^2}{4\Delta t}\right) \tag{26}$$

where $d$ is the dimensionality of the system. The probability of the entire discretized path $\{x_0, x_1, \ldots, x_N\}$ is given by the product of these conditional probabilities:

$$P(x_0, x_1, \ldots, x_N) = P(x_0) \prod_{i=0}^{N-1} P(x_{i+1}|x_i) \tag{27}$$

$$\propto \prod_{i=0}^{N-1} \exp\left(-\frac{(x_{i+1} - x_i + \Delta t \nabla U(x_i))^2}{4\Delta t}\right) \tag{28}$$

$$= \exp\left(-\sum_{i=0}^{N-1} \frac{(x_{i+1} - x_i + \Delta t \nabla U(x_i))^2}{4\Delta t}\right) \tag{29}$$

Taking the limit as $\Delta t \to 0$ and $N \to \infty$ with $N\Delta t = T$ fixed, this sum approaches a path integral:

$$\lim_{\substack{\Delta t \to 0 \\ N\Delta t = T}} \sum_{i=0}^{N-1} \frac{(x_{i+1} - x_i + \Delta t \nabla U(x_i))^2}{4\Delta t} = \frac{1}{4} \int_0^T \|\dot{x}(t) + \nabla U(x(t))\|^2 dt \tag{30}$$

Thus, the path probability can be written as:

$$\mathbb{P}[x(t)] \propto \exp(-\frac{1}{4} \int_0^T \|\dot{x}(t) + \nabla U(x(t))\|^2 dt) \tag{31}$$

The exponent term is recognized as the Onsager-Machlup action functional:

$$S_{\text{OM}}[x] = \frac{1}{4} \int_0^T \|\dot{x}(t) + \nabla U(x(t))\|^2 dt \tag{32}$$

This action functional penalizes paths that deviate from following the force $-\nabla U(x)$. Maximizing the probability of a transition path (subject to fixed endpoints $x(0) = x_A$ and $x(T) = x_B$) is equivalent to minimizing this action functional.

Expanding the integrand:

$$\|\dot{x}(t) + \nabla U(x(t))\|^2 = \|\dot{x}(t)\|^2 + 2\dot{x}(t) \cdot \nabla U(x(t)) + \|\nabla U(x(t))\|^2 \tag{33}$$

For paths with fixed endpoints $x(0) = x_A$ and $x(T) = x_B$, the cross-term integrates to a constant difference in potential energy:

$$\int_0^T \dot{x}(t) \cdot \nabla U(x(t))dt = \int_0^T \frac{d}{dt}U(x(t))dt = U(x_B) - U(x_A) \tag{34}$$

We can define a modified action functional that, when minimized, is equivalent to minimizing the original Onsager-Machlup functional for fixed endpoints:

$$\tilde{S}[x] = \int_0^T (\|\dot{x}(t)\|^2 + \|\nabla U(x(t))\|^2)dt \tag{35}$$

By the Cauchy-Schwarz inequality:

$$\|\dot{x}(t)\|^2 + \|\nabla U(x(t))\|^2 \geq 2\|\dot{x}(t)\|\|\nabla U(x(t))\| \tag{36}$$

With equality when $\dot{x}(t)$ is parallel or antiparallel to $\nabla U(x(t))$, with the direction determined by whether the path is ascending or descending the energy landscape. This is precisely the defining characteristic of the MEP - a path that follows the potential energy landscape's gradient while avoiding high-energy regions.

When this equality holds, our action simplifies to:

$$\tilde{S}[x] = \int_0^T 2\|\dot{x}(t)\|\|\nabla U(x(t))\|dt \tag{37}$$

The term $\|\dot{x}(t)\|dt$ represents the infinitesimal arc length element $ds_{\text{arc}}$ along the path, allowing us to rewrite the action as a line integral:

$$S_{\text{geo}}[x] = 2\int_\varphi \|\nabla U(x)\|ds_{\text{arc}} \tag{38}$$

where $\varphi$ is the path in configuration space. This geometric form is independent of the path's parameterization and depends solely on the potential energy landscape.

### A.2.2 DISCRETIZATION AND CONNECTION TO ENERGY DIFFERENCES

To compute this geometric action numerically, we introduce a progress coordinate $s \in [0, 1]$ and a mapping $\varphi_\theta(s)$ such that $\varphi_\theta(0) = x_A$ and $\varphi_\theta(1) = x_B$, where $\theta$ represents the network parameters. The relationship between the progress coordinate and arc length is:

$$ds_{\text{arc}} = \left\|\frac{d\varphi_\theta}{ds}\right\| ds \tag{39}$$

We discretize the path with $K$ points, setting $s_k = \frac{k}{K-1}$ and $\varphi_k = \varphi_\theta(s_k)$ for $k = 0, 1, ..., K-1$. The discretized geometric action becomes:

$$S_{\text{geo}}[\varphi_\theta] \approx 2\sum_{k=0}^{K-2} \|\nabla U(\varphi_k)\| \cdot \|\Delta\varphi_k\| \tag{40}$$

where $\Delta\varphi_k = \varphi_{k+1} - \varphi_k$.

Using Taylor's theorem, we can relate energy differences to gradient magnitudes:

$$U(\varphi_{k+1}) - U(\varphi_k) = \nabla U(\varphi_k) \cdot \Delta\varphi_k + O(\|\Delta\varphi_k\|^2) \tag{41}$$

$$= \|\nabla U(\varphi_k)\|\|\Delta\varphi_k\|\cos\theta_k + O(\|\Delta\varphi_k\|^2) \tag{42}$$

where $\theta_k$ is the angle between the gradient and displacement vectors.

Critically, this Taylor expansion approximation is only valid when $\|\Delta\varphi_k\|$ is sufficiently small for the higher-order term $O(\|\Delta\varphi_k\|^2)$ to be negligible. When displacement vectors become large, the approximation breaks down. It can lead to pathological solutions, where the optimization finds paths that avoid high-energy transition regions by rapidly "teleporting" between stable states, rather than discovering smooth, minimum-energy routes.

For a path following the minimum-energy trajectory, the gradient aligns with or against the path direction ($\cos\theta_k \to \pm 1$ as $\Delta s \to 0$, depending on whether the path is ascending or descending the energy landscape), giving at the minimum of the optimization:

$$|U(\varphi_{k+1}) - U(\varphi_k)| \approx \|\nabla U(\varphi_k)\| \cdot \|\Delta\varphi_k\| \tag{43}$$

This yields an approximation of the geometric action:

$$S_{\text{geo}}[\varphi_\theta] \approx 2\sum_{k=0}^{K-2} |U(\varphi_{k+1}) - U(\varphi_k)| \tag{44}$$

This forms a surrogate objective with the same minimum as our original action, provided the Taylor approximation remains valid throughout optimization.

### A.2.3 ESTABLISHING THE ENERGY-BASED LOSS FUNCTION

We need to bound the sum of absolute energy differences to establish a direct connection to an energy-based loss function. Using the triangle inequality, for any two points and any constant $c$:

$$|U(\varphi_{k+1}) - U(\varphi_k)| \leq \frac{(U(\varphi_{k+1}) - c) + (U(\varphi_k) - c)}{2} = \frac{U(\varphi_{k+1}) + U(\varphi_k) - 2c}{2} \tag{45}$$

Choosing $c = \min_{s \in [0,1]} U(\varphi_\theta(s))$ to be the minimum energy along the entire path and summing over all segments:

$$\sum_{k=0}^{K-2} |U(\varphi_{k+1}) - U(\varphi_k)| \leq \sum_{k=0}^{K-2} \frac{U(\varphi_{k+1}) + U(\varphi_k) - 2c}{2} \tag{46}$$

$$= \frac{1}{2}\left(\sum_{k=0}^{K-2} U(\varphi_{k+1}) + \sum_{k=0}^{K-2} U(\varphi_k) - 2(K-1)c\right) \tag{47}$$

We can rewrite the first sum as $\sum_{k=1}^{K-1} U(\varphi_k)$ and the second as $\sum_{k=0}^{K-2} U(\varphi_k)$. Combining these terms:

$$\sum_{k=0}^{K-2} |U(\varphi_{k+1}) - U(\varphi_k)| \leq \frac{1}{2}\left(\sum_{k=1}^{K-1} U(\varphi_k) + \sum_{k=0}^{K-2} U(\varphi_k) - 2(K-1)c\right) \tag{48}$$

$$= \frac{1}{2}\left(\sum_{k=0}^{K-1} U(\varphi_k) - U(\varphi_0) + \sum_{k=0}^{K-1} U(\varphi_k) - U(\varphi_{K-1}) - 2(K-1)c\right) \tag{49}$$

$$= \sum_{k=0}^{K-1} U(\varphi_k) - \frac{U(\varphi_0) + U(\varphi_{K-1})}{2} - (K-1)c \tag{50}$$

Since $c$, $U(\varphi_0)$, and $U(\varphi_{K-1})$ are constants during optimization, minimizing $\sum_{k=0}^{K-1} U(\varphi_k)$ is equivalent to minimizing the upper bound on the geometric action. This provides the theoretical justification for our energy-based loss function.

### A.2.4 NEURAL NETWORK IMPLEMENTATION

For practical optimization, we parametrize the path using a neural network $\varphi_\theta(s)$ with parameters $\theta$. To implement our approach, we sample points along the path and minimize the energy at these

sampled locations. This represents a transition from the theoretical derivation using discrete segments to a practical implementation using sampled points for more efficient optimization. We employ stochastic sampling of the progress coordinate to construct our loss function:

$$\mathcal{L}(\theta) = \frac{1}{B} \sum_{j=1}^{B} U(\varphi_\theta(s_j)) \tag{51}$$

where $s_j \sim \mathcal{U}[0, 1]$ are uniformly sampled progress coordinates and $B$ is the batch size (the number of points sampled along the path during each optimization step).

This stochastic approach provides an unbiased estimator of the expected energy along the path:

$$\mathbb{E}_{s \sim \mathcal{U}[0,1]}[U(\varphi_\theta(s))] = \int_0^1 U(\varphi_\theta(s))ds \tag{52}$$

By minimizing $\mathcal{L}(\theta)$ through gradient-based optimization, the neural network learns to represent low-energy pathways connecting the stable states, effectively discovering the MEP predicted by the Onsager-Machlup framework. The resulting path follows the potential energy landscape's gradient while avoiding high-energy regions, representing the most probable transition mechanism in the overdamped limit. However, as noted above, the validity of this approach depends critically on maintaining sufficiently small displacement vectors $\|\Delta\varphi_k\|$ during optimization to ensure the underlying Taylor approximation remains valid.

## B    BENCHMARKS

### B.1    SCOPE OF BENCHMARK COMPARISONS

Our benchmarks focus on methods that directly parameterize transition paths—either as single MEPs or transition path ensembles—using explicit geometric representations. This includes classical discrete methods (Chain-of-States, String Method, NEB) and neural continuous parameterizations (AdaPath, Doob's Lagrangian (Du et al., 2024), Doob's Seq2Seq (Lee et al., 2025)). We exclude approaches that learn biasing potentials or control policies (Yan et al., 2022; Holdijk et al., 2023; Seong et al., 2025) rather than explicit path representations, as these methods require different evaluation frameworks (policy rollouts vs. direct path assessment) and cannot be evaluated using our energy-based metrics without reimplementing their training infrastructure. For Doob's Seq2Seq, we report results on alanine dipeptide as published, but cannot extend the comparison to AIB9 and BPTI due to the lack of released code.

Several recent approaches related to transition path discovery are discussed in Section B.6 but are not included in the quantitative benchmarks for well-defined methodological reasons. Raja et al. (2025) employs a pretrained diffusion model both as a surrogate molecular force field and as an initial path generator. While conceptually related, this formulation replaces classical energy and force evaluations with neural network inference, whose computational cost per evaluation is several orders of magnitude higher and therefore not comparable on a per-force-evaluation basis. Moreover, the resulting potential energy values stem from the learned diffusion surrogate and are not directly comparable to those computed from established molecular force fields. Similarly, Ramakrishnan et al. (2025) introduce a complex implicit neural representation (INR) formulation with a higher-order composite loss involving energies, forces, and path derivatives. We include their reported results to contextualize AdaPath within the broader landscape of emerging neural approaches. However, as no public code is available for their method on atomic systems, a direct 'apples-to-apples' baseline comparison of the full framework is impractical. Nevertheless, to evaluate the efficacy of their optimization objective, their primary contribution, we re-implemented their NEB-inspired loss function. In Section D.5, we benchmark this loss in conjunction with our AdaPath architecture, allowing us to isolate and assess the performance of their training objective against ours.

This scope ensures that all included methods share a comparable objective and computational basis: discovering a continuous minimum-energy path between fixed molecular states using physically consistent molecular force fields. Ensemble-based, beyond Doob's Lagrangian, and surrogate-force-field methods pursue broader or orthogonal objectives that fall outside this scope and are therefore analyzed qualitatively, but not benchmarked quantitatively.

## B.2 BENCHMARK DESIGN AND EVALUATION PROTOCOLS

All benchmarks follow a consistent evaluation protocol to ensure comparability across molecular systems and methods.

**Objective and Metrics:** Each method aims to discover a transition path connecting two fixed endpoint configurations, $x_A$ and $x_B$. Performance is quantified by three complementary metrics:

1. **Max Energy:** The maximum potential energy along the discovered path (in kJ/mol), corresponding to the transition-state energy barrier.

2. **MinMax Energy:** The lowest maximum energy achieved across all random seeds, representing the best discovered MEP.

3. **Mean Energy:** The arc-length-adjusted mean energy along the path, which penalizes methods that traverse high-energy regions too quickly.

Additionally, we report the total number of force field or energy function evaluations (# Evals), providing a direct measure of computational efficiency.

**Benchmark Procedure:** Each benchmark consists of the following stages:

- **System preparation:** Endpoint configurations $x_A$ and $x_B$ are obtained from reference MD trajectories or literature datasets and aligned using the Kabsch algorithm to remove translational and rotational degrees of freedom.

- **Force field consistency:** All methods use identical classical molecular force fields implemented in DMFF to ensure fair energy comparisons.

- **Hyperparameter optimization:** For classical methods, extensive sweeps were conducted over image counts, spring constants, and learning rates to identify optimal settings. Neural methods used identical optimizer and sampling settings unless otherwise noted.

- **Evaluation:** Energies are computed at uniformly spaced points along the path. For stochastic methods, five independent runs are performed with different random seeds, and mean ± standard deviation is reported.

**Computational Environment:** All experiments were conducted on a single NVIDIA A6000 GPU (48 GB memory). Classical methods were implemented in JAX to ensure consistency in energy evaluation and parallelization efficiency.

**Considerations:** While neural ensemble approaches such as Doob's Lagrangian optimize over transition distributions rather than single paths, their reported transition-state energies remain comparable. However, direct comparisons in metrics like trajectory probability or ensemble entropy are omitted, as these quantities are undefined for single-path objectives.

## B.3 BENCHMARKS: ALANINE DIPEPTIDE

We provide a direct comparison between our AdaPath method, Du et al. (2024), Lee et al. (2025), as well as classical, well-established methods like the string method, chain-of-states, and nudged elastic band, in terms of computational efficiency, as evaluated by the number of force field/energy function calls, and the maximum energies observed along the paths, corresponding to the peak of the potential barrier, both as a mean across five seeds as well as the lowest value achieved across all runs. We further compute the arc-length-adjusted mean energy along the path. We chose this metric because naive averaging along a discretized path does not account for how quickly a path traverses a high-energy region; it could, therefore, shortcut a region of high energy but have a very low non-adjusted energy.

To provide a rigorous comparison, we conducted benchmarks using experimental conditions identical to those reported by Du et al. for alanine dipeptide: AMBER14 force field, 300 K temperature, and also referred to their results when applicable, referencing their results with adjusted significant figures for consistency. Table 4 presents these results alongside those reported in Du et al. (2024), with all

it's variants including cartesian and internal coordinate parameterizations, and their recent follow-up work Lee et al. (2025). The only exception is that we do not evaluate trajectory probability, as the assumption of optimization over a non-physical path length makes calculating this quantity ill-defined. Furthermore, the arc-length-adjusted energy quantities are not applicable to Doob's method, as they do not model a single path but rather the entire transition path ensemble.

For the classical methods, we conducted extensive hyperparameter sweeps to ensure fair comparison and optimal performance. Chain-of-States was optimized across path discretizations (20–60 images), spring constants ($10^{-7}$–$10^{-1}$), and learning rates ($5 \times 10^{-2}$–$10^{-1}$), with the best configuration using 40 images, a spring constant of $10^{-6}$, and the Adam optimizer with a learning rate of $5 \times 10^{-2}$. The String Method was evaluated across image counts (20–60), timesteps (0.1–4000), and reparameterization frequencies (1–10 steps), achieving optimal performance with 40 images, timestep 3900, and reparameterization every three steps. NEB was tested across various path discretizations (20–60 images), spring constants ($10^{-7}$–$10^{-1}$), and learning rates ($10^{-6}$–$10^{-1}$), with the best results obtained using 60 images, a spring constant of $2 \times 10^{-5}$, and the Adam optimizer with a learning rate of $10^{-5}$. These configurations represent the optimal settings identified through systematic exploration of the hyperparameter space for each method. All classical approaches we study are deterministic and hence are only run once, unlike the stochastic approaches, which are run five times.

Since the field lacks standardized benchmarks, we believe that building on the work from (Du et al., 2024) could be a helpful step toward having a more standardized approach to methods aiming to solve for MEPs, the transition path ensemble, or other similar approaches. We want to reiterate that (Du et al., 2024) is more ambitious in its sampling objectives, and therefore, one-to-one comparisons are not entirely appropriate. However, the transition state energy, as represented by the Max Energy entry, should be consistent between MEP approaches and methods that aim to sample the entire transition ensemble.

AdaPath requires approximately $3.8\times$ fewer energy function evaluations (6.4e4 vs. 2.4e5) compared to the best classical method (Chain-of-States) while achieving superior energy barriers ($-86.6$ vs. $-71.1$ kJ/mol for the minimum maximum energy). Compared to the Doob's Lagrangian approach, AdaPath requires approximately $800\times$ fewer energy function evaluations (6.4e4 vs. 5.1e7) while achieving significantly lower maximum energy barriers. The classical methods demonstrate substantial variation in computational efficiency, with Chain-of-States providing the best computational performance among baseline approaches in terms of function evaluations. However, AdaPath still requires fewer evaluations while achieving better energy minimization. The lower number of required evaluations for AdaPath is likely due to the efficient neural representation compared to discrete path sampling approaches used by classical methods, as well as the large number of images that these methods require to function well.

Table 4: Comprehensive benchmark comparison of transition path methods on alanine dipeptide system. We distinguish baselines by how the endpoints are defined: via Collective Variables (CV), Relaxed energy states, or Exact coordinates. Max Energy represents the mean $\pm$ standard deviation across five seeds. MinMax Energy shows the lowest maximum energy achieved in any single run.

| Method | # Evals | Max Energy | MinMax Energy | Mean Energy |
|---|---|---|---|---|
| *MCMC Baselines (CV States)* | | | | |
| MCMC variable length | 2.1e7 | $740 \pm 700$ | 52.4 | N/A |
| MCMC* | 1.3e9 | $288 \pm 128$ | 60.5 | N/A |
| *MCMC Baselines (Relaxed States)* | | | | |
| MCMC variable length | 1.9e8 | $413 \pm 335$ | 27.0 | N/A |
| MCMC | $> 1e10$ | N/A | N/A | N/A |
| *MCMC Baselines (Exact States)* | | | | |
| MCMC variable length | $> 1e10$ | N/A | N/A | N/A |
| MCMC | $> 1e10$ | N/A | N/A | N/A |
| *Doob's Lagrangian (Exact States)* | | | | |
| Doob's Cartesian | 3.8e7 | $726.40 \pm 0.07$ | 726.2 | N/A |
| w/ 2 Mixtures | 5.1e7 | $709 \pm 162$ | 513.7 | N/A |
| w/ 5 Mixtures | 5.1e7 | $541 \pm 278$ | 248.0 | N/A |
| Doob's Internal | 3.8e7 | $-14.62 \pm 0.02$ | $-14.67$ | N/A |
| w/ 2 Mixtures | 5.1e7 | $-15.38 \pm 0.14$ | $-15.54$ | N/A |
| w/ 5 Mixtures | 5.1e7 | $-15.5 \pm 0.3$ | $-15.95$ | N/A |
| *Other Neural Baselines* | | | | |
| Doob's Seq2Seq | – | $3.46 \pm 0.03$ | 3.46 | N/A |
| MaxLL | – | 233 | 233 | N/A |
| *Classical Methods (Exact States)* | | | | |
| Chain-of-States | 2.4e5 | $-71.1$ | $-71.1$ | $-82.9$ |
| String Method | 3.3e6 | $-67.7$ | $-67.7$ | $-85.3$ |
| NEB | 7.7e6 | $-67.7$ | $-67.7$ | $-83.5$ |
| **AdaPath (Exact States)** | 6.4e4 | $-80.26 \pm 0.20$ | $-86.58$ | $-85.1 \pm 1.6$ |

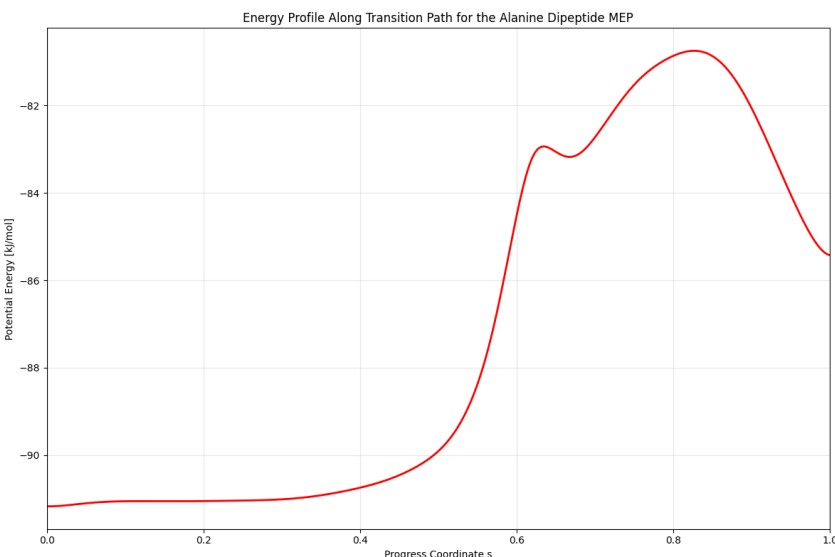

Figure 4: Energy profile along the discovered alanine dipeptide transition pathway. The plot shows the potential energy landscape as a function of the progress coordinate $s \in [0, 1]$ for one representative MEP connecting the $\alpha_R$ and $C_{7ax}$ conformational states. Despite its simplicity as a benchmark system, the profile reveals the characteristic energy barrier governing the conformational transition.

## B.4 Benchmarks: AIB9

We conducted similar benchmarks on the AIB9 peptide system, a 129-atom artificial protein that exhibits well-defined metastable states and realistic conformational changes. This system serves as an intermediate complexity benchmark between simple dipeptides and larger protein systems. For the classical methods, we performed comprehensive hyperparameter optimization: Chain-of-States was evaluated across path discretizations (20 to 60 images), spring constants (1e-6 to 1e-1), and learning rates (1e-3 to 1e-1), achieving optimal performance with 40 images, spring constant 1e-5, and Adam optimizer with learning rate 1e-2. The String Method was tested across various image counts (20 to 60), timesteps (1 to 20), and reparameterization frequencies (1 to 15 steps), yielding the best results with 60 images, a timestep of 10, and reparameterization every three steps. NEB was optimized across path discretizations (20 to 60 images), spring constants (1e-7 to 1e-1), and learning rates (1e-6 to 1e-1), yielding an optimal configuration with 60 images, a spring constant of 1e-3, and the Adam optimizer with a learning rate of 1e-5.

For the Doob's Lagrangian method, we adopted the settings reported for the Chignolin system in Du et al. (2024), as it represents a protein system of similar size. Due to observed training instability, we added gradient clipping (1e-2) to stabilize optimization. We also found the original parameterization to be slightly under-parameterized for this system and increased the MLP width to 768 per layer across five layers. While the resulting energies appear high, they are consistent with the 3000 kJ/mol values reported for Chignolin in the original work, suggesting this energy scale is characteristic of the method's behavior on protein systems of this size.

Table 5: Performance comparison of classical transition path methods, Doob's Lagrangian, and AdaPath on AIB9 peptide system. Results are presented in terms of computational efficiency, energy barriers (kJ/mol), and path quality metrics. All methods were optimized through systematic hyperparameter sweeps to ensure fair comparison. Max Energy represents the mean $\pm$ standard deviation across multiple seeds for neural methods, and the best single result for classical methods. MinMax Energy shows the lowest maximum energy barrier achieved across all runs. Mean Energy accounts for path traversal speed through different energy regions (kJ/mol).

| Method | # Evals | Max Energy | MinMax Energy | Mean Energy |
|---|---|---|---|---|
| Chain-of-States | 2.9e6 | $-461.0$ | $-461.0$ | $-687.2$ |
| String Method | 3.1e6 | 1.3e4 | 1.3e4 | 3.2e3 |
| NEB | 4.1e6 | 2.7e3 | 2.7e3 | $-128.0$ |
| Doob's Lagrangian | 1.2e7 | $1.6e4 \pm 7.8e3$ | 5.8e3 | N/A |
| AdaPath | 4.7e6 | $-496 \pm 179$ | $-644$ | $-767 \pm 24$ |

The results demonstrate AdaPath's superior performance compared to both classical methods and the Doob's Lagrangian approach on this complex system. AdaPath achieved the best overall energy minimization, with a minimum maximum energy of $-644$ kJ/mol, substantially outperforming Chain-of-States ($-461$ kJ/mol), String Method (1.3e4 kJ/mol), NEB (2.7e3 kJ/mol), and Doob's Lagrangian (5.8e3 kJ/mol). The arc-length-adjusted mean energies further highlight AdaPath's advantages, achieving $-767 \pm 24$ kJ/mol. Notably, AdaPath demonstrated both superior energy minimization and significantly improved consistency across multiple seeds, with 3 out of 5 seeds achieving excellent results below $-590$ kJ/mol, while Doob's Lagrangian showed high variability with a standard deviation of 7.8e3 kJ/mol. The substantial energy differences between methods highlight the importance of the optimization objective and neural architecture design for effective transition path discovery.

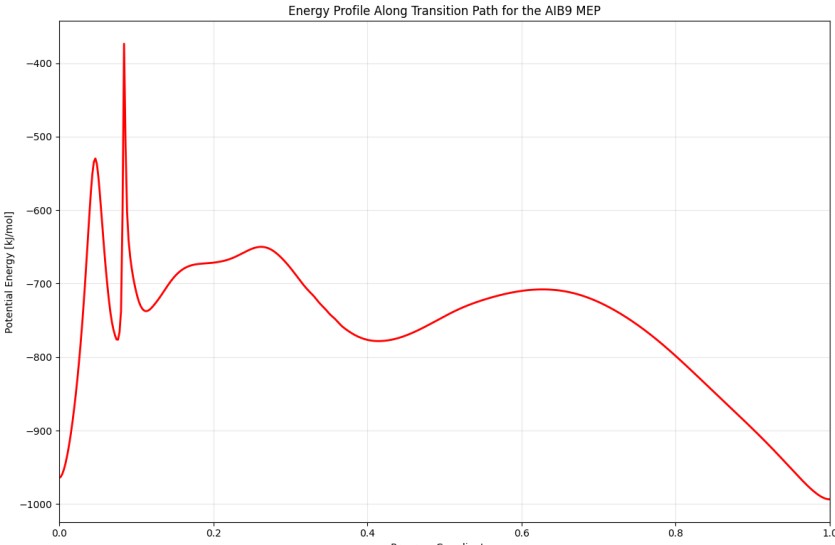

Figure 5: Energy profile along the discovered AIB9 transition pathway. The plot shows the potential energy landscape as a function of the progress coordinate $s \in [0, 1]$ for one representative MEP connecting the two metastable conformational states of the 129-atom peptide. The two sharp spikes might indicate suboptimal solutions at these parts of the parameterization.

## B.5 BENCHMARKS: BPTI

We conducted benchmarks on the BPTI (Bovine Pancreatic Trypsin Inhibitor) system to evaluate our AdaPath method against classical transition path discovery approaches on a significantly larger and more complex biomolecular system, BPTI. The system setup, including explicit solvation, ionic strength, and force field parameters, is detailed in Appendix F.

For the classical methods (Chain-of-States, String Method, and NEB), we employed the same hyperparameter configurations that proved optimal for the AIB9 system rather than conducting comprehensive sweeps, as the computational cost of extensive hyperparameter optimization on a system of this size would be prohibitive. Specifically, we used the following methods: Chain-of-States with 40 images, a spring constant of 1e-5, and the Adam optimizer with a learning rate of 1e-2; String Method with 60 images, a timestep of 10, and reparameterization every three steps; and NEB with 60 images, a spring constant of 1e-3, and the Adam optimizer with a learning rate of 1e-5.

For AdaPath, we used the training parameters specified in Appendix F, which is a more parameter-efficient variant of the model that performed best on the smaller AIB9 system.

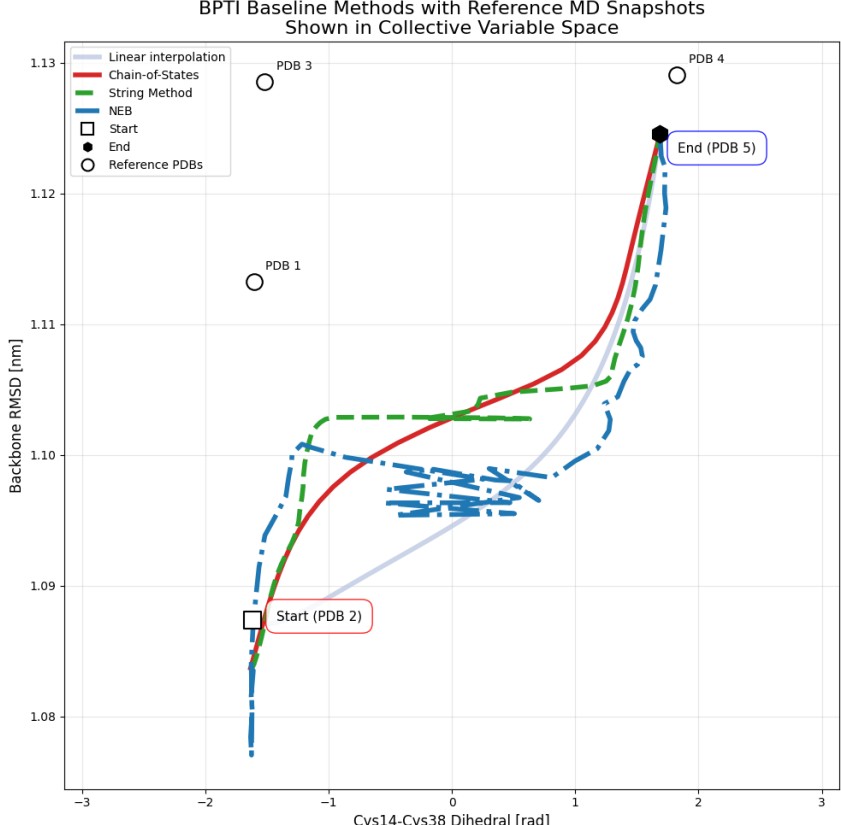

Figure 6: Transition paths discovered by classical methods for the BPTI system projected onto the same collective variables used in the main paper: Cys14-Cys38 disulfide torsion angle versus backbone RMSD of residues 4-54. The Chain-of-States method (red solid line), String Method (green dashed line), and NEB (blue dash-dot line) either fail to converge and/or show erratic zigzagging behavior. Reference structures from the D.E. Shaw MD simulation are shown as open circles (PDB 1, 3, 4), with the start configuration (PDB 2) and end configuration (PDB 5) marked accordingly. Notably, none of the classical methods successfully navigate through all intermediate reference structures, in contrast to AdaPath's performance shown in the main paper.

Table 6: Performance comparison of classical transition path methods and AdaPath on the BPTI system with explicit solvent. Results show computational efficiency, energy barriers (kJ/mol), and path quality metrics. Max Energy shows mean $\pm$ std dev across seeds for AdaPath, best single result for classical methods.

| Method | # Evals | Max Energy | MinMax Energy | Mean Energy |
|---|---|---|---|---|
| Chain-of-States | 1.4e6 | $-1.3e4$ | $-1.3e4$ | $-2.0e4$ |
| String Method | 3.8e6 | 1.4e4 | 1.4e4 | 4.3e3 |
| NEB | 1.1e6 | 1.6e5 | 1.6e5 | 2.3e5 |
| AdaPath | 9.6e4 | $-7.1e4 \pm 4.1e4$ | $-1.4e5$ | $-2.2e5 \pm 1.9e5$ |

The results demonstrate AdaPath's superior performance on this complex, explicitly solvated system. Figure 6 shows that among classical approaches, Chain-of-States produces the most reasonable pathway but fails to navigate through all intermediate reference structures from the D.E. Shaw simulation. The String Method fails to converge to a good solution, while NEB displays a jagged path, a known issue for this method in high-dimensional systems (Zhang et al., 2016). In contrast,

AdaPath successfully discovers transition paths that pass through all intermediate reference structures, as demonstrated in Figure 3, while achieving significantly lower maximum energy barriers.

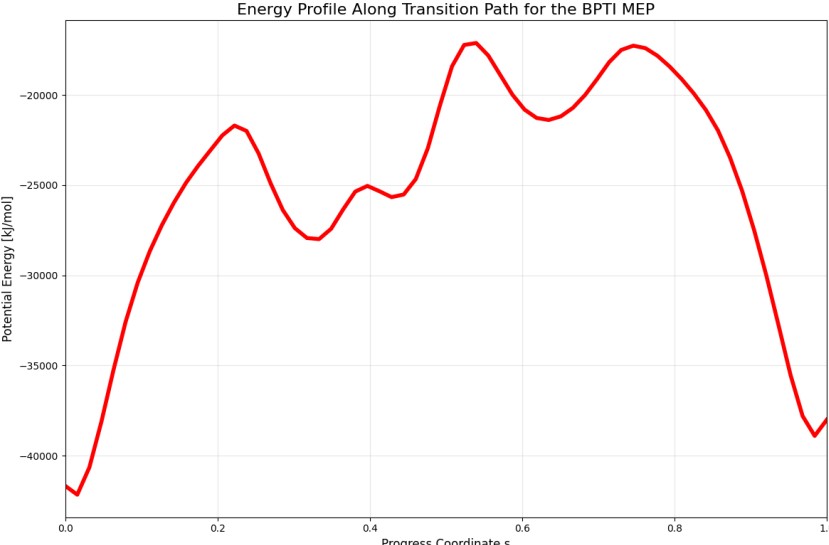

Figure 7: Energy profile along the discovered BPTI transition pathway. The plot shows the potential energy landscape as a function of the progress coordinate $s \in [0, 1]$ for one representative MEP connecting the two BPTI conformational states. The profile reveals multiple intermediate energy barriers and metastable states along the transition pathway.

### B.6 OTHER RELATED METHODS

While our primary benchmark comparison focuses on the Doob's Lagrangian method and classical MEP approaches, due to similarity in task and system setup, examining how our approach relates to other methods for transition path discovery/MEP generation is also informative. Here, we present results from two related approaches in the literature. However, we emphasize that direct numerical comparisons should be treated with caution due to differences in force fields, optimization objectives, and units used.

Table 7: Comparison between nudged elastic band (NEB) and implicit neural representation (INR) approaches for alanine dipeptide, adapted from Ramakrishnan et al. (2025). The table shows the transition state (TS) energies achieved by each method after their complete optimization procedure, alongside the total number of force field evaluations required.

| Method | # Evaluations | Max Energy (eV) |
| --- | --- | --- |
| NEB | 8.1e3 | 11.0 |
| INR | 3.0e3 | 0.407 |

Table 7 presents results from Ramakrishnan et al. (2025) comparing the Nudged Elastic Band (NEB) method with their Implicit Neural Representation (INR) approach for MEP discovery. Both our AdaPath approach and their INR method utilize lightweight, preexisting molecular dynamics force fields, though we employ different such force fields. Their approach requires a more complex loss function that evaluates both energy and forces along with path parameterization and its gradient (path velocity). However, it needs fewer force field evaluations than our method. It is important to note that the "Max Energy" reported in Table 7 is measured in electron volts (eV) and appears to be adjusted for ground state energy reference, which makes direct energy comparisons with our method challenging. The difference in force fields used further complicates any direct energy value comparisons between methods.

Table 8: Computational efficiency comparison between different sampling methods for transition paths on alanine dipeptide, adapted from Raja et al. (2025). The table compares traditional methods with an Onsager-Machlup (OM) optimization approach that uses a pre-trained diffusion model as a surrogate for molecular force fields. The "CVs" column indicates whether collective variables were required (a potential limitation).

| Method | CVs | # Evaluations | Runtime / Path |
|---|---|---|---|
| MCMC (Two-Way Shooting) | No | $\geq$ 1e9 | $\geq$ 100 hours |
| Metadynamics | Yes | 1e6 | 10 hours |
| OM Opt. (Diffusion Model) | No | 1e4 | 50 min |

In contrast, Table 8 presents results from Raja et al. (2025), who take a fundamentally different approach by repurposing pre-trained generative models to perform sampling of transition paths. A key advantage of their method is that the generative model provides significantly better initial guesses for transition paths based on learned molecular distributions. Their approach requires evaluating both the pretrained model (as a surrogate force field) and its gradient for their loss function, which is more computationally intense due to the high number of parameters of the pretrained model as compared to a physics-derived force field.

These tables corroborate our findings in the Doob's Lagrangian comparison: methods that optimize for a single MEP consistently require orders of magnitude fewer force field evaluations than approaches that characterize the complete transition path ensemble.

### B.7 PATH VELOCITY ANALYSIS

To assess whether our energy-based loss function leads to pathological "overstretching" behavior where paths rapidly traverse transition regions, we compute velocity metrics along discovered transition paths. Path velocity is defined as the spatial derivative magnitude $v(s) = \|\frac{d\varphi(s)}{ds}\|$, which measures how quickly the molecular configuration changes with respect to the progress coordinate.

For neural methods, velocities are computed via finite differences:

$$v(s_i) \approx \frac{\|\varphi(s_i + \epsilon) - \varphi(s_i - \epsilon)\|}{2\epsilon} \tag{53}$$

where $\epsilon = 10^{-5}$ and evaluations are performed at 64 uniformly spaced points along $s \in [0, 1]$.

For classical discrete methods, velocities are computed between adjacent images in the discretized path:

$$v_i = \frac{\|\varphi_{i+1} - \varphi_i\|}{\Delta s} \tag{54}$$

where $\Delta s = 1/(N - 1)$ for $N$ images.

**Interpretation of Velocity Metrics:** Since all methods parameterize paths using a dimensionless progress coordinate $s \in [0, 1]$ rather than physical time, absolute velocity magnitudes alone do not indicate pathological behavior. A longer path through conformational space naturally exhibits higher velocities because the spatial displacement per unit progress is larger. The critical diagnostic for pathological "teleportation" behavior is the ratio between maximum and mean velocities: extreme overstretching manifests as localized discontinuous jumps, producing very large max/mean velocity ratios where the path suddenly accelerates through transition regions. In contrast, smoothly parameterized paths maintain relatively uniform velocities with modest max/mean ratios. Tables 9, 10, and 11 present velocity comparisons across our benchmark systems.

Table 9: Path velocity comparison for Alanine dipeptide system. The max/mean velocity ratio serves as the primary diagnostic for pathological overstretching, with values near 1.0 indicating uniform parameterization. For AdaPath, results show mean $\pm$ standard deviation across five independent runs.

| Method | Mean Velocity (nm) | Max Velocity (nm) | Max/Mean Ratio |
|---|---|---|---|
| Chain-of-States | 1.12 | 3.54 | 3.17 |
| String Method | 1.41 | 1.41 | 1.0 |
| NEB | 4.74 | 6.70 | 1.42 |
| AdaPath | $0.96 \pm 0.03$ | $1.4 \pm 0.6$ | $1.4 \pm 0.6$ |

Table 10: Path velocity comparison for AIB9 peptide system. Velocity ratios demonstrate that AdaPath maintains smooth path parameterizations comparable to classical methods without exhibiting pathological overstretching behavior.

| Method | Mean Velocity (nm) | Max Velocity (nm) | Max/Mean Ratio |
|---|---|---|---|
| Chain-of-States | 7.1 | 7.5 | 1.05 |
| String Method | 13.0 | 13.0 | 1.0 |
| NEB | 38.3 | 41.4 | 1.08 |
| AdaPath | $8.6 \pm 0.2$ | $11.9 \pm 1.5$ | $1.38 \pm 0.20$ |

Table 11: Path velocity comparison for BPTI system with explicit solvent. Results on this large-scale system further validate that the energy-based loss function maintains physically reasonable path parameterizations without extreme velocity variations.

| Method | Mean Velocity (nm) | Max Velocity (nm) | Max/Mean Ratio |
|---|---|---|---|
| Chain-of-States | 21 | 27 | 1.3 |
| String Method | 21 | 21 | 1.0 |
| NEB | 48 | 79 | 1.7 |
| AdaPath | $100 \pm 70$ | $180 \pm 140$ | $1.7 \pm 0.3$ |

Across all three systems, AdaPath maintains max/mean velocity ratios comparable to classical methods, with values ranging from $1.384 \pm 0.197$ for AIB9 to $1.689 \pm 0.261$ for BPTI. These modest ratios indicate smooth, uniformly parameterized paths without the extreme velocity spikes characteristic of pathological teleportation behavior. The String Method achieves the most uniform parameterization (ratio $\approx 1.0$) due to its explicit arc-length reparameterization. Notably, AdaPath's ratios remain well within the range of established classical methods, confirming that our energy-based optimization does not produce discontinuous or overstretched transition paths while achieving superior energy barriers as reported in Sections B.3, B.4, and B.5.

B.8  WALL-CLOCK PERFORMANCE BENCHMARKS

To provide a comprehensive assessment of computational efficiency, we supplement our force evaluation counts with wall-clock timing measurements. This analysis accounts for the actual time required to discover minimum-energy paths rather than solely counting operations, providing a more practical metric for comparing method efficiency.

**Benchmark Methodology:**  For each method, we measure the wall-clock time per training step using the following protocol:

1. **JIT Compilation Warmup:** We perform 5 warmup iterations to ensure all JAX functions are fully JIT-compiled and GPU kernels are loaded into memory. These warmup iterations are excluded from timing measurements.

2. **Statistical Sampling:** We measure 50 independent training steps, recording the wall-clock time for each iteration using Python's `time.perf_counter()`. Each measurement includes gradient computation, parameter updates, and JAX's `block_until_ready()` to ensure asynchronous operations complete.

3. **Total Time Calculation:** We calculate the total wall-clock time required for each method's complete optimization run. This is computed as: `total_time = time_per_step ×` `n_iterations`, where `n_iterations` is the number of training steps, until the best maximum energy was reached, used in our benchmark experiments reported in Appendices B.3, B.4, and B.5.

4. **Hardware Consistency:** All timing measurements were conducted on a single NVIDIA A6000 GPU with 48GB memory under identical system load conditions.

**Results:** Tables 12, 13, and 14 present wall-clock timing results for all classical methods as well as ours, on the three molecular systems. The results demonstrate that while our method achieves superior energy minimization, the wall-clock performance varies by system size.

Table 12: Wall-clock performance comparison on alanine dipeptide system. Time per step represents the mean wall-clock time for a single training iteration after JIT compilation warmup. Total time reflects the complete optimization run using the iteration counts, until the best maximum energy was achieved, from Appendix B.3.

| Method | Iterations | Time/Step (ms) | Total Time (min) | # Evals |
|---|---|---|---|---|
| Chain-of-States | 6,000 | 0.29 | 0.03 | 2.4e5 |
| String Method | 82,500 | 0.32 | 0.44 | 3.3e6 |
| NEB | 128,333 | 0.38 | 0.82 | 7.7e6 |
| AdaPath | 4,000 | 3.5 | 0.24 | 6.4e4 |

Table 13: Wall-clock performance comparison on AIB9 peptide system. Results show the computational time required for the complete optimization runs reported in Appendix B.4.

| Method | Iterations | Time/Step (ms) | Total Time (min) | # Evals |
|---|---|---|---|---|
| Chain-of-States | 72,500 | 0.8 | 0.9 | 2.9e6 |
| String Method | 51,667 | 0.8 | 0.7 | 3.1e6 |
| NEB | 68,333 | 0.9 | 1.0 | 4.1e6 |
| AdaPath | 293,750 | 3.9 | 18.9 | 4.7e6 |

Table 14: Wall-clock performance comparison on BPTI system with explicit solvent. The larger system size (3,500 atoms) demonstrates different scaling characteristics for each method.

| Method | Iterations | Time/Step (ms) | Total Time (min) | # Evals |
|---|---|---|---|---|
| Chain-of-States | 35,000 | 19 | 10.8 | 1.4e6 |
| String Method | 63,333 | 27 | 28.0 | 3.8e6 |
| NEB | 18,333 | 27 | 8.2 | 1.1e6 |
| AdaPath | 6,000 | 91 | 9.1 | 9.6e4 |

The wall-clock analysis reveals system-size-dependent performance characteristics. On the smallest system (alanine dipeptide), classical methods complete faster in absolute time due to their simpler per-step operations, though AdaPath achieves superior energy minimization with fewer total evaluations. On AIB9, AdaPath requires longer wall-clock time but discovers significantly better MEPs (minimum maximum energy of $-644$ kJ/mol vs. Chain-of-States' $-461$ kJ/mol). On BPTI, AdaPath's wall-clock performance becomes competitive with classical methods while maintaining its advantage

in path quality, requiring only 96,000 force evaluations compared to over 1 million for classical approaches. The per-step timing reflects the different computational profiles: classical methods have very low per-step overhead but require many more iterations, while AdaPath performs more computation per step through neural network forward and backward passes but converges with dramatically fewer force evaluations on large systems.

## C    CONNECTION BETWEEN DOOB'S LAGRANGIAN AND MEP OPTIMIZATION

We elaborate on the method of Doob's h-transform for sampling transition paths (Du et al., 2024). This explanation is relevant as both methods employ neural networks to represent molecular state transitions, either as a minimum-energy path in our case or the complete transition path ensemble in theirs. Notably, the parameterization of the mean trajectory in Du et al. resembles our MEP representation, differing primarily in our use of blending functions and spatial embeddings for the neural network component.

Doob's Lagrangian formulation considers the following action functional:

$$S = \min_{q,v} \int_0^T dt \int dx \, q_{t|0,T}(x) \langle v_{t|0,T}(x), G_t v_{t|0,T}(x) \rangle \tag{55}$$

This functional quantifies the cost of controlling a stochastic process to achieve desired endpoint conditions, subject to the system's energy landscape. Intuitively, it measures the amount of "effort"/action needed to steer the dynamics from starting state A to target state B, with smaller values indicating more probable transition paths.

In this formulation, $q_{t|0,T}(x)$ is the probability density at time $t$ given boundary conditions, parameterized as a Gaussian $\mathcal{N}(x|\mu_{t|0,T}, \Sigma_{t|0,T})$, and $v_{t|0,T}(x)$ is the control vector field satisfying $v_{t|0,T}(x) = \frac{1}{2}(G_t)^{-1}(u_{t|0,T}(x) - b_t(x))$, with $b_t(x) = -\nabla V(x)$ representing the reference drift for overdamped dynamics.

This action functional describes a Wasserstein Lagrangian flow as formalized in (Neklyudov et al., 2023) and (Neklyudov et al., 2024). In this framework, one can learn stochastic dynamics from samples, thereby learning a process that defines a time-dependent density evolving from an initial to a final state.

The difference with our approach is that we minimize the action of a single deterministic path rather than a complete distribution of paths. This simplification is reflected in our loss function, which contains one fewer expectation than the full Doob's Lagrangian.

## D    ABLATION STUDIES

This section presents a series of ablation studies examining the impact of various components of our AdaPath approach. These experiments help validate our design choices and provide insights into the importance of different architectural and training elements. We conduct these ablation studies on the AIB9 system because Alanine dipeptide, as a benchmark system, is too simple to provide meaningful insights into the performance of our method. For each ablation study, we report the lowest maximum energy of the path among all training runs, the mean and standard deviation of the arc-length adjusted energy, and the mean number of iterations to reach the lowest maximum energy. We run each setting in the ablations 5 times. We use a batch size of 1, where each batch contains 16 values of the progress coordinate $s$ for most cases.

### D.1    GENERAL SETTINGS FOR ABLATION STUDIES

All ablation studies are conducted using the following base configuration, with each study varying only specific parameters while keeping all others fixed:

**Molecular System Parameters:**

- Molecular system: AIB9 peptide with 129 atoms

- Force field: AMBER ff15ipq-m for protein mimetics

- Cutoff method: No cutoff applied

- Simulation box size: 10.0 nm

- Neighbor list cutoff distance: 4.0 nm

**Training Configuration:**

- Number of iterations: 100,000 per run

- Number of independent runs: 5 per parameter setting

- Evaluation frequency: Every 2000 training steps

- Optimizer: Muon as implemented in Optax.

- Learning rate: 1e-2

**Path Sampling Parameters:**

- Batch size: 1

- Frames per batch: 16 points sampled along the progress coordinate $s$

- Interpolation method: Linear

**Network Architecture (Base Configuration):**

- Network depth: 6 MLP blocks.

- MLP Expansion Factor: 4, meaning that each MLP block, consisting of two linear layers, has four times the hidden dimensionality as input and output dimensionality.

- Atom Embedding Dimension: 32.

- Time embedding: Sinusoidal with maximum period of 1

- Activation function: GeLu throughout the network

- Conditioning MLP settings: 3 Layers, with a hidden dimension of 1024 and GeLu activations.

## D.2    ARCHITECTURE COMPARISON: ADAPATH VS STANDARD MLP

To validate our core claim that parameter sharing across atoms provides superior scalability, we compare AdaPath against a standard MLP baseline that directly maps progress coordinate $s$ to the complete molecular configuration without parameter sharing. This comparison is conducted across all three molecular systems (Alanine dipeptide, AIB9, and BPTI) to demonstrate scaling behavior.

The standard MLP baseline uses a conventional feedforward architecture with six hidden layers of width determined by 4 times the number of atoms in the system. This was found to be suitable for the two small systems. Intuitively, this aligns well with deep learning's collective wisdom of having a higher hidden dimension than the input and output dimensions of the network. We only lower this to one time the atom count for the BPTI system, as this already leads to an MLP with 1= billion parameters, as more was not computationally infeasible for our hardware. The network takes sinusoidal embeddings of the progress coordinate $s$ as input and directly outputs the $3N$-dimensional molecular configuration, where $N$ is the number of atoms in the system. The standard MLP (TransitionMLP) uses the same interpolation strategy and endpoint constraints as AdaPath but processes all atomic coordinates through shared dense layers rather than using per-atom embeddings with parameter sharing.

All other training parameters remain identical to the base configuration described in Section D.1, with both architectures using the same learning rate of 1e-2 with the Muon optimizer for the two smaller systems and the Nadamw optimizer for BPTI as outlined in F.

Table 15: Architecture comparison between AdaPath and standard MLP on Alanine dipeptide system. Results demonstrate the parameter efficiency and performance of the parameter-sharing approach on this simple benchmark system.

| Architecture | Parameters | Max Energy | MinMax Energy | Mean Energy |
|---|---|---|---|---|
| Standard MLP | 5e4 | $5 \pm 160$ | $-68.1$ | $-25 \pm 103$ |
| AdaPath | 2e6 | $-80.3 \pm 0.2$ | $-86.6$ | $-85.1 \pm 1.6$ |

Table 16: Architecture comparison between AdaPath and standard MLP on AIB9 peptide system. The 129-atom system provides a more challenging test case for evaluating architectural scaling behavior.

| Architecture | Parameters | Max Energy | MinMax Energy | Mean Energy |
|---|---|---|---|---|
| Standard MLP | 2e6 | $4.3e4 \pm 9.3e3$ | $2.99e4$ | $1.92e4 \pm 2.3e3$ |
| AdaPath | 2e6 | $-496 \pm 179$ | $-644$ | $-767 \pm 24$ |

Table 17: Architecture comparison between AdaPath and standard MLP on the BPTI system with explicit solvent. This large system provides the most stringent test of architectural scalability claims.

| Architecture | Parameters | Max Energy | MinMax Energy | Mean Energy |
|---|---|---|---|---|
| Standard MLP | 1.4e9 | $2.2e4 \pm 2.0e4$ | $3.0e3$ | $1.9e4 \pm 1.2e4$ |
| AdaPath | 3.5e6 | $-7.1e4 \pm 4.1e4$ | $-1.4e5$ | $-2.2e5 \pm 1.9e5$ |

The results demonstrate AdaPath's superior performance and scalability across system sizes. The AdaPath's parameter count is dominated by the progress coordinate MLP that outputs the adaLn parameters. This produces a baseline parameter count that puts the AdaPath architecture above the MLP for small systems, but for larger ones, this gap inverts sharply.

On the alanine dipeptide system, AdaPath achieves significantly lower and more consistent maximum energies (-86.6 vs -68.1 kJ/mol for minimum maximum energy) with substantially reduced variance. The performance gap becomes dramatic on the larger AIB9 system, where the standard MLP achieves a best maximum energy of 2.99e4 kJ/mol compared to AdaPath's -644 kJ/mol, representing a 30,000+ kJ/mol improvement in path quality.

### D.3 OPTIMIZER COMPARISON: PATH CONTINUITY ANALYSIS

To validate our claim that the Muon optimizer maintains better path continuity compared to Adam for small systems, we conduct a comparative analysis using path velocity metrics over training iterations. We train identical AdaPath networks on the AIB9 system using both optimizers with learning rates of 1e-2 for Muon and 1e-4 for Adam, respectively. All other training parameters remain identical to the base configuration.

We track the maximum path velocity over training iterations, computed as $\max_{s \in [0,1]} \|\frac{d\varphi_\theta(s)}{ds}\|$, where high velocities indicate rapid changes in molecular configuration that can lead to discontinuous "teleportation" between states rather than smooth transitions.

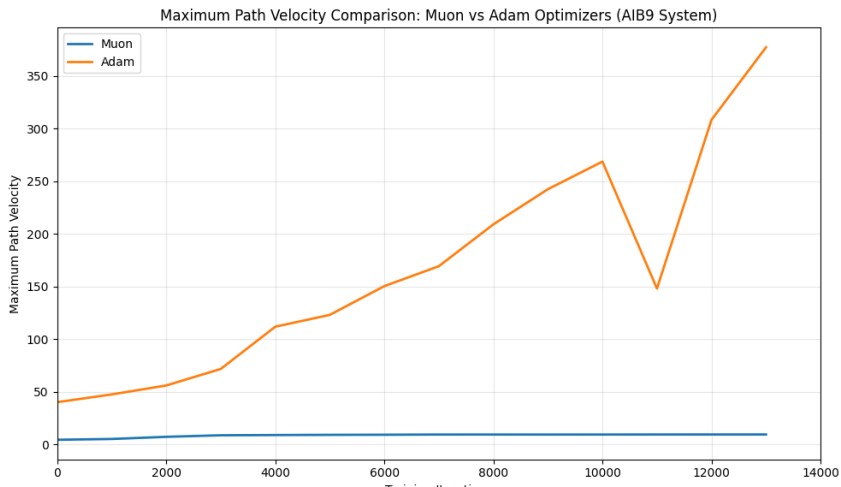

Figure 8: Maximum path velocity over training iterations comparing Muon and Adam optimizers on the AIB9 system. Path velocity is computed as the maximum spatial derivative along the transition path at each training step. High velocities indicate rapid configurational changes that can lead to discontinuous transitions, while moderate velocities suggest smoother, more physically meaningful pathways.

Table 18: Optimizer comparison between optimizers on AIB9 system.

| Optimizer | Max Energy | Mean Energy |
|---|---|---|
| Adam | $3.0e3$ | $-802$ |
| Muon | $-496 \pm 179$ | $-767 \pm 24$ |

The Adam-trained system exhibits extremely high path velocities, indicating that it rapidly switches from one state to another without passing through the transition region. This is corroborated by the low mean but high max energies.

## D.4 ENERGY TRANSFORMATION ANALYSIS

We evaluate the impact of our logarithmic energy transformation on training stability by comparing performance with and without the log transformation applied to the potential energy in the loss function. The logarithmic transformation is applied as $\mathcal{L}_{\log}(\theta) = \frac{1}{B} \sum_{j=1}^{B} \log(U(\varphi_\theta(s_j)) + c)$ where $c = 0.1$ ensures positivity, while the baseline uses the direct energy $\mathcal{L}_{\text{direct}}(\theta) = \frac{1}{B} \sum_{j=1}^{B} U(\varphi_\theta(s_j))$. All other training parameters remain identical to the base configuration.

Table 19: The impact of logarithmic energy transformation on AIB9 system performance and training stability.

| Energy Transform | Max Energy | MinMax Energy | Mean Energy |
|---|---|---|---|
| Direct Energy | $3.3e4 \pm 1.7e4$ | $3.0e3$ | $6.4e3 \pm 3.1e3$ |
| Log Transform | $-496 \pm 179$ | $-644$ | $-767 \pm 24$ |

The logarithmic transformation achieves significantly lower energies with reduced variance, demonstrating improved training stability and path quality over direct energy optimization.

### D.5 LOSS FUNCTION COMPARISON: ADAPATH VS. INR

To validate our loss function design, we compare our energy-based approach against the composite loss formulation from Ramakrishnan et al. (2025), which represents paths using implicit neural representations (INRs) with NEB-inspired terms. The INR approach uses a three-term loss function:

$$\mathcal{L}_{\text{INR}}(\theta) = \mathbb{E}_t[\tilde{U}(\varphi_\theta(t))] + \lambda_s \text{Var}_t[\|\varphi_\theta'(t)\|] - \lambda_{\text{cl}}\text{Stop}[\nabla U(\varphi_\theta(t^*))_\|]^T \varphi_\theta(t^*) \tag{56}$$

where the modified energy removes tangential force components. The first term minimizes path energy while discarding tangential gradients (nudging), the second promotes uniform path velocity ($\lambda_s$), and the third implements climbing-image behavior ($\lambda_{\text{cl}}$) by pushing the highest-energy sample up the potential surface. This formulation requires computing path derivatives, tangent projections, and selective gradient masking, adding computational overhead compared to our direct energy minimization.

We conduct hyperparameter sweeps on all three systems, optimizing over learning rates $\{10^{-2}, 10^{-3}, 10^{-4}, 10^{-5}, 10^{-6}\}$ and batch sizes $\{16, 32, 64, 128\}$. Following Ramakrishnan et al. (2025) for molecular systems, we fix $\lambda_s = 0$ and $\lambda_{\text{cl}} = 1.0$. For BPTI, we transfer the best AIB9 hyperparameters to avoid prohibitive computational cost. All other settings follow Appendix D.1.

The INR loss requires 2.2 x per training step on alanine dipeptide compared to our energy-based loss due to path derivative calculations.

Table 20: Loss function comparison on alanine dipeptide system. INR uses LR=$10^{-2}$, n_frames=32. Results show mean $\pm$ standard deviation across five independent runs.

| Method | # Evals | Max Energy | MinMax Energy | Mean Energy |
|--------|---------|------------|---------------|-------------|
| INR | 1.1e6 | $-80.54 \pm 0.33$ | $-81.02$ | $-83.9 \pm 0.8$ |
| AdaPath | 6.4e4 | $-80.26 \pm 0.20$ | $-86.58$ | $-85.1 \pm 1.6$ |

Table 21: Loss function comparison on AIB9 peptide system. Best INR results use LR=$10^{-5}$, batch size=64. Results show mean $\pm$ standard deviation across three independent runs.

| Method | # Evals | Max Energy | MinMax Energy | Mean Energy |
|--------|---------|------------|---------------|-------------|
| INR | 5.1e6 | $743 \pm 830$ | $-394$ | $398 \pm 206$ |
| AdaPath | 4.7e6 | $-496 \pm 179$ | $-644$ | $-767 \pm 24$ |

Table 22: Loss function comparison on BPTI system with explicit solvent. Hyperparameters transferred from AIB9 experiments (LR=$10^{-5}$, batch size=64). Results show mean $\pm$ standard deviation across five independent runs.

| Method | # Evals | Max Energy | MinMax Energy | Mean Energy |
|--------|---------|------------|---------------|-------------|
| INR | 5.0e5 | $-2.4e4 \pm 0.6e4$ | $-3.2e4$ | $-4.7e4 \pm 2.0e4$ |
| AdaPath | 9.6e4 | $-7.1e4 \pm 4.1e4$ | $-1.4e5$ | $-2.2e5 \pm 1.9e5$ |

The INR loss exhibits instability on AIB9. While the best run achieved -394 kJ/mol (MinMax), the method was unstable with a mean barrier of $743 \pm 830$ kJ/mol. This large variance ($\pm 830$ kJ/mol) compared to our energy-based approach ($\pm 179$ kJ/mol) might indicate optimization challenges with the multi-term loss formulation. On the larger BPTI system, the INR method achieves a best maximum energy of $-3.2e4$ kJ/mol compared to AdaPath's $-1.4e5$ kJ/mol, representing a substantial gap in path quality. Additionally, the INR approach required approximately 5× more force field evaluations (5.0e5 vs. 9.6e4) to achieve these results.

# E  AIB9 SYSTEM PREPARATION

### E.0.1  REFERENCE TRAJECTORY GENERATION

To generate reference conformational data, we performed an extended molecular dynamics simulation using OpenMM with the following parameters:

- **Force field**: AMBER ff15ipq-m
- **Temperature**: 500 K (elevated to enhance sampling of conformational transitions)
- **Integrator**: Langevin Middle Integrator with friction coefficient of $1 \text{ ps}^{-1}$
- **Time step**: 1 femtosecond
- **Non-bonded treatment**: No cutoff (NoCutoff method)
- **Constraints**: Hydrogen bonds constrained using the HBonds method
- **Total simulation time**: 900 nanoseconds
- **Output frequency**: Coordinates saved every 1000 steps (1 ps intervals)

The simulation protocol consisted of initial energy minimization followed by velocity assignment at the target temperature, a brief 10 ps equilibration period, and then the production run. The elevated temperature of 500 K was chosen to accelerate conformational sampling and ensure adequate transitions between metastable states during the simulation timeframe.

### E.0.2  STATE DEFINITION AND ANALYSIS

To characterize the conformational landscape, we focused on the backbone dihedral angles of the central residues. Specifically, we computed the $\phi$ and $\psi$ dihedral angles for residues at indices [56, 58, 60, 69] and [58, 60, 69, 71], respectively, corresponding to the central amino acids of the peptide chain.

Two metastable states were defined using ellipsoidal boundaries in the $\phi$-$\psi$ dihedral space:

$$\text{State A}: \quad \left(\frac{\phi + 1.0}{0.1}\right)^2 + \left(\frac{\psi + 0.45}{0.1}\right)^2 \leq 1 \tag{57}$$

$$\text{State B}: \quad \left(\frac{\phi - 1.0}{0.1}\right)^2 + \left(\frac{\psi - 0.45}{0.1}\right)^2 \leq 1 \tag{58}$$

These elliptical regions capture the distinct conformational basins observed in the free energy landscape projected onto the central residue dihedral angles.

### E.0.3  NEURAL NETWORK TRAINING SETUP

For the neural network-based MEP optimization, we employed a different force field setup optimized for differentiable molecular dynamics:

- **Force field**: AMBER ff15ipq-m for protein mimetics, implemented through DMFF
- **Non-bonded treatment**: No cutoff method for consistency with the reference simulation
- **Simulation box**: 11.0 nm cubic box
- **Neighbor list cutoff**: 5.0 nm

### E.0.4  START- AND END-STATE SETUP

Start and end configurations for MEP optimization were randomly sampled from the reference trajectory frames that satisfied the respective state definitions. The Kabsch algorithm was employed to optimally align the end configuration to the start configuration, removing translational and rotational degrees of freedom. This alignment procedure ensures that the neural network focuses on learning the internal conformational changes rather than rigid-body motions, leading to more efficient and physically meaningful transition path discovery.

## F    BPTI System Preparation

As reference conformations, we used the structures from the D.E. Shaw Research BPTI simulation of BPTI (Shaw et al., 2010). The initial and final states for our MEP optimization were selected as the second snapshot (index 1) and the second-to-last snapshot (index -2) from the provided reference structures, as they were the most separated in the CVs determined in the original study, named the backbone RMSD and the disulfide torsion angle.

### F.1    Explicit Solvent System Preparation

For the BPTI system, we developed an approach to handle explicit water molecules and ions using periodic boundary conditions:

- **System solvation**: Both start and end protein configurations were placed in 3.5 nm cubic periodic boxes and solvated with TIP3P water molecules using OpenMM's Modeller class. The system included $Na^+$ and $Cl^-$ ions at 0.04 M ionic strength to approximate physiological conditions.

- **Energy minimization**: To preserve the reference protein conformations from the D.E. Shaw simulation while allowing solvent relaxation, all protein heavy atoms were fixed by setting their masses to zero during energy minimization. This completely freezes the protein structure while allowing water molecules and ions to relax.

- **Molecule matching**: Since the endpoint configurations were solvated independently, we equalized atom counts by randomly removing excess molecules, then used the Hungarian algorithm to optimally pair water molecules and ions between configurations based on center-of-mass positions. This matching minimizes solvent displacement during transitions.

The resulting system contained approximately 3,500 atoms total in a periodic cubic box, maintaining proper solvation while using standard periodic boundary conditions compatible with conventional MD force fields.

### F.2    Training Parameters

**Architecture and Training Parameters:**

- MLP Expansion Factor: 2
- Layers: 5
- Hidden dimension: 64
- Progress embedding dimension: 1024
- Conditioning depth: 3 layers
- Conditioning hidden size: 1024
- Optimizer: NAdam with weight decay (nadamw)
- Learning rate: $1 \times 10^{-4}$
- Gradient clipping threshold: $1 \times 10^{-1}$
- Weight decay: $1 \times 10^{-3}$
- Beta parameters: $\beta_1 = 0.9$, $\beta_2 = 0.99$, $\epsilon = 1 \times 10^{-8}$
- Frames per batch: 16
- Total iterations: 6,000
- Evaluation frequency: Every 1000 iterations

### F.3    Force Field Parameters

We used the following force field parameters to match the D.E. Shaw simulation conditions:

- **Protein force field**: AMBER99sb-ILDN (amber99sbildn.xml)
- **Water force field**: TIP3P (tip3p.xml)

## G   VILLIN HP35 SYSTEM PREPARATION

We additionally examine the villin headpiece subdomain (HP35), a 35-residue fast-folding protein that serves as a well-established benchmark system in the protein folding literature. Villin HP35 has been extensively characterized through both experimental studies and computational simulations, including in the work of Pina et al. (Piana et al., 2012), which extensively simulated the system and multiple mutants. The availability of reference simulation data and the system's intermediate complexity, larger than simple dipeptides but smaller than the 58-residue BPTI, makes it an ideal validation case for assessing whether our discovered MEPs align with independently derived free energy landscapes. This provides an additional check on the physical relevance of our transition pathways beyond the BPTI comparison with D.E. Shaw simulation snapshots.

### G.0.1   REFERENCE TRAJECTORY AND STATE DEFINITION

We follow the settings of (Piana et al., 2012) for the setup of the MD system, and use their data as both the source of the start and end points, as well as as reference. **Trajectory Data:**

- **Source**: Multiple DCD trajectory files from enhanced sampling simulations at 345 K

- **Topology**: Villin HP35 (35 residues, 583 atoms)

- **Force field**: AMBER99SB-ILDN

**Torsion-Angle TICA Analysis:**

To define metastable states, we performed TICA on all backbone dihedral angles ($\phi$, $\psi$, $\omega$) and sidechain angles ($\chi_1 - \chi_5$):

1. All dihedral angles were computed using MDTraj and converted to sine/cosine pairs: $[\sin\theta, \cos\theta]$ for each angle $\theta$

2. TICA was performed with lag time $\tau = 50$ ps (100 frames) and 2 output dimensions with kinetic map scaling

3. States were defined as circular regions in TICA space:

$$\text{State A (folded)}: \quad \text{TICA} - [1.6, -0.42] \leq 0.1 \tag{59}$$
$$\text{State B (unfolded)}: \quad \text{TICA} - [-1.75, -3.4] \leq 0.1 \tag{60}$$

Representative structures were selected as frames closest to each state center in TICA space.

### G.0.2   EXPLICIT SOLVENT SYSTEM PREPARATION

Both start and end configurations were prepared with explicit solvation, matching the source papers' conditions:

- **System solvation**: Protein configurations placed in 3.0 nm cubic periodic boxes and solvated with TIP3P water using OpenMM's Modeller class. $Na^+$ and $Cl^-$ ions added at 0.04 M ionic strength.

- **Energy minimization**: All protein heavy atoms were fixed by setting masses to zero during minimization. This preserved reference conformations while allowing water and ion relaxation.

- **Molecule matching**: After independent solvation, atom counts were equalized by removing excess molecules. The Hungarian algorithm optimally paired water molecules and ions between configurations based on center-of-mass positions.

The resulting system contained 2,188 atoms total (583 protein + 1,602 water + 3 $Cl^-$ ions).

### G.0.3 TRAINING PARAMETERS

**Architecture:**

- Hidden dimension: 32
- MLP layers: 6
- Expansion factor: 2
- Activation: ReLU
- Normalization: Layernorm
- Dropout: 0.0
- Output scale: 0.0
- Progress embedding: Fourier features with max period 1.0

**Training Configuration:**

- Optimizer: Adam
- Learning rate: $1 \times 10^{-4}$
- Frames per batch: 16
- Neighbor list cutoff: 1.0 nm
- Neighbor list update frequency: Every 100 iterations

### G.0.4 RESULTS

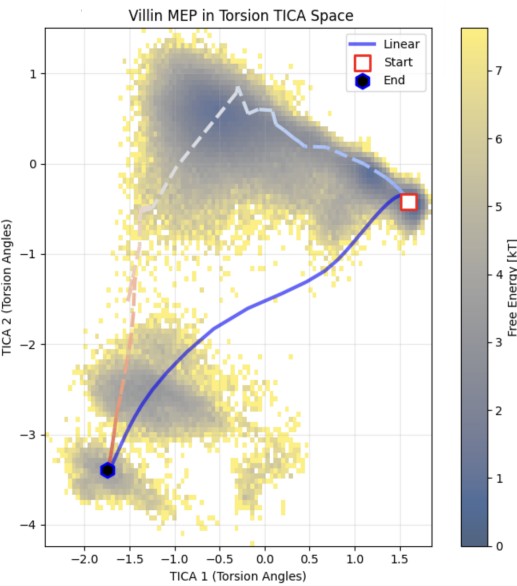

Figure 9: Villin HP35 MEP in torsion-angle TICA space. Background: free energy landscape from reference MD. Dashed red-to-blue line: optimized neural MEP (gradient coloring shows progress from start to end). Solid blue line: linear interpolation. The neural MEP follows low free energy valleys, thereby avoiding high-barrier regions that are traversed by the linear baseline.

Figure 9 shows the optimized MEP projected onto the TICA free energy landscape. The background free energy surface $F(x) = -k_B T \ln P(x)$ was computed from the reference trajectory.

The discovered MEP passes through an intermediate region.

## G.1 COMMITTOR ANALYSIS ALONG DISCOVERED MEPS

To validate the physical relevance of our discovered minimum-energy paths, we performed committor analysis on intermediate configurations along the MEPs for both alanine dipeptide and AIB9 systems. The committor function $p_B(x)$ represents the probability that a trajectory initiated from configuration $x$ with velocities sampled from the Maxwell-Boltzmann distribution will reach state B before state A. Configurations with $p_B \approx 0.5$ lie on the transition state ensemble and represent the true dynamical bottleneck for the conformational transition.

### G.1.1 COMMITTOR ANALYSIS METHODOLOGY

For each point along the MEP, we computed the committor probability using the following protocol:

1. **Configuration sampling:** We uniformly sampled 11 configurations along the discovered MEP by evaluating the neural network at progress coordinates $s \in [0, 1]$ for alanine dipeptide, and 61 configurations along $s \in [0.2, 0.7]$ for AIB9.

2. **Trajectory generation:** For each configuration, we initiated multiple independent MD trajectories using OpenMM with Langevin dynamics. Initial velocities were drawn from the Maxwell-Boltzmann distribution at the target temperature.

3. **Trajectory classification:** Each trajectory was propagated until it reached either state A or state B, or until a maximum simulation time was reached. States were defined using the same dihedral angle criteria employed for endpoint selection.

4. **Committor estimation:** The committor probability $p_B$ at each configuration was estimated as $p_B = N_B/(N_A + N_B)$, where $N_A$ and $N_B$ are the numbers of trajectories reaching states A and B, respectively.

### G.1.2 SYSTEM-SPECIFIC PARAMETERS

**Alanine Dipeptide:**

- **Temperature:** 300 K
- **Maximum trajectory length:** 20 ps
- **Trajectories per configuration:** 100

**AIB9:**

- **Temperature:** 500 K (elevated to enhance sampling)
- **Maximum trajectory length:** 30 ps
- **Trajectories per configuration:** 30

### G.1.3 ANALYTICAL COMMITTOR FROM THE MEP ENERGY PROFILE

While empirical committor calculations via trajectory shooting provide ground-truth validation, they are computationally expensive. Here, we recapitulate the derivation in (L. Pontryagin et al., 1933) of an analytical expression for the committor based on the one-dimensional energy profile, in our case along the MEP, $\varphi(s)$, enabling rapid estimation of committor values directly from the discovered pathway.

**Setup:** Consider the MEP $\varphi : [0, 1] \to \mathbb{R}^{3N}$ with an energy landscape $U(\varphi(s))$ along the path. We assume overdamped Langevin dynamics at temperature $T$. The committor $q(s)$ is the probability that a trajectory starting at $\varphi(s)$ reaches $x_B$ before $x_A$, subject to boundary conditions $q(s_A) = 0$ and $q(s_B) = 1$, where $s_A$ and $s_B$ denote the progress coordinates at which the path exits state A and enters state B, respectively.

**Governing Equation:** For overdamped dynamics, the committor satisfies the steady-state Backward Fokker-Planck equation:

$$\frac{d}{ds}\left(e^{-\beta U(\varphi(s))}\frac{dq}{ds}\right) = 0 \tag{61}$$

where $\beta = 1/(k_B T)$.

**Analytical Solution:** Since the derivative of the term in parentheses vanishes, we have $e^{-\beta U(\varphi(s))}\frac{dq}{ds} = C$ for some constant $C$. Rearranging and integrating from $s_A$ to $s$:

$$q(s) = C\int_{s_A}^{s} e^{\beta U(\varphi(s'))}\,ds' \tag{62}$$

Applying the boundary condition $q(s_B) = 1$ determines the constant, yielding:

$$q(s) = \frac{\int_{s_A}^{s} e^{\beta U(\varphi(s'))}\,ds'}{\int_{s_A}^{s_B} e^{\beta U(\varphi(s'))}\,ds'} \tag{63}$$

**Physical Interpretation:** The integrand $e^{\beta U(\varphi(s))}$ is large when the potential energy is high, representing the "resistance" to crossing a specific region of the path. For a symmetric barrier. At low temperature (high $\beta$), $e^{\beta U}$ becomes very large at the barrier top and negligible elsewhere, causing $q(s)$ to exhibit step-function behavior.

**Numerical Implementation:** For numerical stability with large energy barriers, we implement Equation 63 using log-sum-exp operations. The state boundaries $s_A$ and $s_B$ are detected by evaluating the state classifier functions along the MEP.

### G.1.4    RESULTS AND INTERPRETATION

Figure 10 presents the committor probability as a function of MEP progress coordinate for both systems, comparing the empirical committor from trajectory shooting with the analytical committor derived from the one-dimensional energy profile.

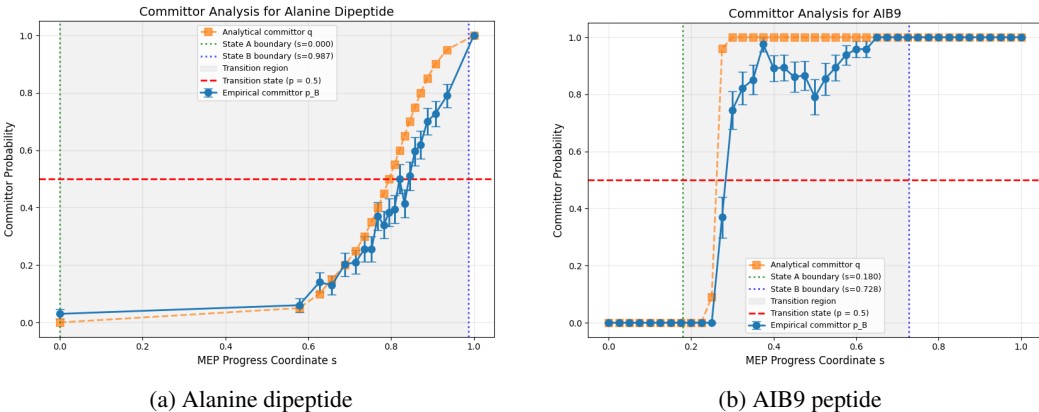

(a) Alanine dipeptide                         (b) AIB9 peptide

Figure 10: Committor analysis comparing empirical (blue circles) and analytical (orange squares) approaches for (a) alanine dipeptide and (b) AIB9 peptide. The analytical committor $q$ is derived from the one-dimensional energy profile along the MEP using Equation 63. Both methods identify similar transition regions.

The MEP combined with the analytical committor provides a systematic way to identify configurations near the transition state ($q \approx 0.5$) without extensive trial-and-error shooting. By inverting Equation 63, one can sample MEP coordinates corresponding to target committor values, enabling "committor-guided" sampling where configurations uniformly span the committor range.

## H  TRAINING STABILITY

Atomistic force fields present unique optimization challenges due to the presence of Lennard-Jones interactions, which are typically modeled using the 12-6 potential:

$$V_{\text{LJ}}(r_{ij}) = 4\epsilon_{ij} \left[ \left( \frac{\sigma_{ij}}{r_{ij}} \right)^{12} - \left( \frac{\sigma_{ij}}{r_{ij}} \right)^{6} \right] \tag{64}$$

where $r_{ij}$ is the distance between atoms $i$ and $j$, $\epsilon_{ij}$ is the well depth, and $\sigma_{ij}$ is the collision diameter. The $r^{-12}$ repulsive term creates extremely steep energy barriers when atoms approach each other too closely, leading to numerical instabilities during neural network training. As $r_{ij} \to 0$, the potential energy diverges to infinity, causing gradient magnitudes that can destabilize the optimization process.

To mitigate these issues, we employ a two-pronged approach. First, we apply a logarithmic transformation to the potential energy after shifting it to ensure positivity:

$$\mathcal{L}_{\text{transformed}}(\theta) = \frac{1}{B} \sum_{j=1}^{B} \log(U(\phi_\theta(s_j)) + c) \tag{65}$$

where $c$ is a positive constant to ensure the argument of the logarithm remains positive. This transformation preserves the location of energy minima while significantly compressing the magnitude of high-energy spikes. The monotonic nature of the logarithm ensures that $\arg\min_\theta \mathcal{L}_{\text{transformed}}(\theta) = \arg\min_\theta \mathcal{L}(\theta)$, preserving the optimization objective. However, the model now places different weights on different parts of the path than before. In practice, we find that this drastically improves training stability.

Second, we implement gradient norm-based gradient clipping to handle any remaining numerical instabilities. This combination of logarithmic energy transformation and gradient clipping provides robust training stability while maintaining the physical validity of the discovered minimum-energy paths.

### H.1  LOGARITHMIC ENERGY TRANSFORMATION PERTURBATION ANALYSIS

Consider the true minimum energy path $\varphi^*(s)$ for $s \in [0, 1]$ satisfying:

- Boundary conditions: $\varphi^*(0) = x_A$ and $\varphi^*(1) = x_B$
- Orthogonality condition: $[\nabla U(\varphi^*(s))]^\perp = \nabla U - (\nabla U \cdot \hat{\tau})\hat{\tau} = 0$

where $\hat{\tau} = \frac{d\varphi^*/ds}{|d\varphi^*/ds|}$ is the unit tangent vector along the path.

Consider a perturbed path $\varphi(s) = \varphi^*(s) + \epsilon\delta(s)$ where $\epsilon \ll 1$ and $\delta(0) = \delta(1) = 0$ (fixed endpoints). At each point, decompose the perturbation:

$$\delta(s) = \delta_\parallel(s) + \delta_\perp(s) \tag{66}$$

where $\delta_\parallel = (\delta \cdot \hat{\tau})\hat{\tau}$ and $\delta_\perp \cdot \hat{\tau} = 0$.

Under perturbation, the energy at each point becomes:

$$U(\varphi^* + \epsilon\delta) = U(\varphi^*) + \epsilon\nabla U(\varphi^*) \cdot \delta + \frac{\epsilon^2}{2}\delta^T H \delta + O(\epsilon^3) \tag{67}$$

where $H$ is the Hessian of $U$, and we examine the perpendicular components of the perturbation and how they interact with the Hessian, as these are the components of interest in this context. Since $[\nabla U(\varphi^*)]^\perp = 0$ at the MEP:

$$U(\varphi^* + \epsilon\delta) = U(\varphi^*) + \epsilon[\nabla U]_\parallel \cdot \delta_\parallel + \frac{\epsilon^2}{2}\delta_\perp^T H_\perp \delta_\perp + O(\epsilon^3) \tag{68}$$

The change in the discretized original loss:

$$\Delta\mathcal{L} = \frac{1}{B} \sum_{j=1}^{B} [U(\varphi^*(s_j) + \epsilon\delta(s_j)) - U(\varphi^*(s_j))] \tag{69}$$

Substituting the energy expansion:

$$\Delta\mathcal{L} = \frac{\epsilon}{B} \sum_{j=1}^{B} [\nabla U]_{\parallel} \cdot \delta_{\parallel}(s_j) + \frac{\epsilon^2}{2B} \sum_{j=1}^{B} \delta_{\perp}^T(s_j) H_{\perp} \delta_{\perp}(s_j) \tag{70}$$

For the log-transformed loss:

$$\Delta\mathcal{L}_{\log} = \frac{1}{B} \sum_{j=1}^{B} [\log(U(\varphi^* + \epsilon\delta) + c) - \log(U(\varphi^*) + c)] \tag{71}$$

$$= \frac{1}{B} \sum_{j=1}^{B} \log\left(1 + \frac{\epsilon[\nabla U]_{\parallel} \cdot \delta_{\parallel} + \frac{\epsilon^2}{2}\delta_{\perp}^T H_{\perp} \delta_{\perp}}{U(\varphi^*) + c}\right) \tag{72}$$

Using $\log(1 + x) \approx x$ for small $x$:

$$\Delta\mathcal{L}_{\log} = \frac{\epsilon}{B} \sum_{j=1}^{B} \frac{[\nabla U]_{\parallel} \cdot \delta_{\parallel}(s_j)}{U(\varphi^*(s_j)) + c} + \frac{\epsilon^2}{2B} \sum_{j=1}^{B} \frac{\delta_{\perp}^T(s_j) H_{\perp} \delta_{\perp}(s_j)}{U(\varphi^*(s_j)) + c} \tag{73}$$

Both losses exhibit:

1. No first-order response to perpendicular perturbations: The $\delta_{\perp}$ terms appear only at second order

2. Positive-definite second-order response: Since the MEP follows energy valleys, $H_{\perp} > 0$

3. Restoring forces: Both $\Delta\mathcal{L} > 0$ and $\Delta\mathcal{L}_{\log} > 0$ for any $\delta_{\perp} \neq 0$

The difference lies only in the weighting factor $\frac{1}{U(\varphi^*)+c}$, which scales the magnitude but not the direction of the restoring force. Therefore, both optimizations converge to the same geometric path.

### H.1.1 Parameterization and Path Stretching

While both losses converge to the same MEP geometry, they differ in how they parameterize the path. The parallel component of the perturbation affects the path's parameterization speed $v(s) = |d\varphi/ds|$.

The gradient with respect to shifting a sample point along the path:

$$\frac{\partial\mathcal{L}}{\partial s_j} = \nabla U(\varphi(s_j)) \cdot \frac{d\varphi}{ds}\bigg|_{s_j} = |\nabla U|_{\parallel} \cdot v(s_j) \tag{74}$$

This creates forces that push sample points away from high-energy regions, potentially leading to extreme stretching where $v(s) \to \infty$ at transition states.

The corresponding gradient for the log loss:

$$\frac{\partial\mathcal{L}_{\log}}{\partial s_j} = \frac{1}{U(\varphi(s_j)) + c} |\nabla U|_{\parallel} \cdot v(s_j) \tag{75}$$

The weighting factor $\frac{1}{U+c}$ provides implicit regularization:

- At transition states (high $U$): The factor is small, reducing the incentive to stretch
- At stable states (low $U$): The factor is larger but bounded

## I   Computational Performance

For the BPTI system with explicit solvent (approximately 3,500 atoms total):

- **Training time**: Approximately 9 minutes on a single NVIDIA A6000 GPU
- **Force field evaluations**: 96000 evaluations

- **Trajectory generation**: 64 frames for the final MEP visualization and analysis

This computational effort is approximately six orders of magnitude less, in node-hours, than the original D.E. Shaw Research simulation on their specialized Anton supercomputer, which required approximately 1.3 million node-hours to generate the 1.03 millisecond trajectory capturing the same conformational changes.

## J COMPUTATIONAL RESOURCES

All experiments reported in this paper were conducted on a single NVIDIA A6000 GPU with 48GB of memory. Additional computational resources were used during the development phase of this project for hyperparameter tuning across network architectures and optimization parameters.

