# OpenReview forum: "Follow the MEP: Scalable Neural Representations for Minimum-Energy Path Discovery in Molecular Systems"
_ICLR.cc/2026/Conference — Submitted to ICLR 2026_

### Official Review · Reviewer_9Rgq · 2025-10-17

**Soundness:** 3
**Presentation:** 2
**Contribution:** 3
**Rating:** 6
**Confidence:** 4

**Summary:**

This work presents AdaPath, which targets efficient discovery of minimum-energy transition paths for large biomolecular systems. Built upon the Onsager-Machlup action functional, AdaPath adopts a simple energy-based loss, which reformulated the MEP discovery as a continuous neural optimization problem. Meanwhile, the model is established with learnable embedding sharing across atoms and a conditioning network for the progress coordinate, which is tailored for scalability. Experiments on the 9-residue peptide AIB9 and the 58-residue protein BPTI with explicit water demonstrate the scalability of AdaPath to explore transition paths for large biomolecular systems with thousands of atoms.

**Strengths:**

- This paper presents a concise and effective derivation and simplification of the OM action potential, requiring only two endpoints and a given potential function for training. This greatly simplifies the overall training procedure, thereby enhancing the practical utility of AdaPath.
- The model architecture is well designed: it addresses the challenge of continuous paths through interpolation, and the use of shared atom embeddings enhances the model’s scalability.
- AdaPath is able to reasonably sample transition paths in both the AIB9 and BPTI systems, which differ greatly in scale. Moreover, it successfully reproduces intermediate states in the large BPTI system, comprising thousands of atoms with explicit water molecules, which is quite impressive.

**Weaknesses:**

- It is noted that AdaPath embeds only the progress coordinate and atom types, while lacking molecular topology and structural information in its inputs. This implies that the model must be trained separately for each system, making out-of-distribution generalization virtually impossible.

**Questions:**

1. AdaPath uses only atomic embeddings and the progress coordinate as inputs, without incorporating any topological or structural information. This design likely limits its out-of-distribution generalization capability, requiring retraining for each unseen system. Does this pose a challenge to the model’s practical applicability? I would appreciate clarification from the authors on this point.
2. Appendix G.1 shows that AdaPath undergoes a two-stage training process on the BPTI system, with different hyperparameters used at each stage. This raises some concerns regarding the complexity and robustness of the training procedure. If the model were to be applied to other molecular systems, would it require task-specific fine-tuning of the training strategy, or is the current training procedure generally applicable? I would like the authors to provide empirical evidence supporting the robustness of the training process across different systems.
3. Although the experiments on the BPTI system are quite impressive, evaluating the model on only two cases (i.e., AIB9 and BPTI) is not sufficient to demonstrate its overall feasibility. I recommend that the authors include additional examples of transition paths for biologically relevant processes. For instance, using fast-folding proteins [1] as target systems to study transitions between unfolded and folded states.
4. Section 2.2 reads more like a summary of related work, and placing it within the Method section may lead to confusion. It is recommended that the authors present this part as a separate section.

**References**

> [1] Lindorff-Larsen, K., Piana, S., Dror, R. O., & Shaw, D. E. (2011). How fast-folding proteins fold. Science, 334(6055), 517-520.

**Details Of Ethics Concerns:**

None.

---

> ### Author Response · Authors · 2025-11-20
>
> Response to Reviewer 9Rgq
>
> We thank the reviewer for the positive and encouraging evaluation and for recognizing the strengths of our approach.
>
> 1. Concern regarding lack of topological information and generalization
>
> This is indeed a valid point, and we thank the reviewer for raising it. We rewrote the architecture section to more clearly describe how AdaPath processes atom embeddings and why the architecture is per-system. While AdaPath must be retrained for each molecular system, the training time is only a few minutes per system, making this requirement practical.
>
> AdaPath inherits the flexibility of molecular dynamics: users may define arbitrary combinations of proteins, ligands, and endpoint configurations without requiring pretraining. In contrast, more general pretrained approaches, such as those in [2], do not require retraining but still entail substantially longer runtimes for sampling transitions and are limited by their training data. Thus, the per-system retraining cost of AdaPath is modest relative to the sampling cost of these methods while providing more flexibility.
>
> 2. Robustness of the training procedure and two-stage schedule
>
> We appreciate the reviewer’s concern. During the review period, we re-examined our training and found that our gradient clipping threshold had been set too aggressively. This resulted in gradients being clipped almost every iteration, which destroyed Adam’s momentum estimates and caused instability.
>
> After correcting this issue, AdaPath trains stably without requiring a two-stage learning-rate schedule. We updated the BPTI results to reflect this. The method is now substantially more robust.
>
> 3. Request for additional biologically relevant examples
>
> We followed the reviewer’s suggestion and added a fourth system: Villin HP35, a fast-folding protein with extensive MD reference data [3]. The system setup is described in the appendix.
>
> AdaPath’s recovered MEP follows low free-energy valleys and passes through known intermediates from independent folding studies, providing additional evidence that AdaPath recovers physically meaningful mechanisms.
>
> 4. Placement of related work
>
> We reorganized the related-work discussion:
>
> - High-level context appears in the Introduction.
> - Detailed comparisons and benchmark-scope explanations appear in the appendix.
> - Section 2 now focuses exclusively on our formulation and architecture.
>
> This improves clarity and separates background from methodology.
>
> 5. Summary of broader revisions
>
> We also:
>
> - Added committor analyses and velocity diagnostics,
> - Compared AdaPath against the INR loss [1] and other ML baselines,
> - Moved major benchmark tables into the main paper,
> - Strengthened the derivation and structure in Section 2.
>
> These revisions required substantial new experiments and significantly improved both clarity and rigor.
>
> We thank the reviewer again for the constructive feedback. Given these improvements and the additional biological validation, we respectfully ask that you maintain or increase your already positive score.
>
> References
> [1] Ramakrishnan et al. (2025).
> [2] Raja et al. (2025).
> [3] Piana et al. (2012), PNAS.

---

> > ### Comment · Reviewer_9Rgq · 2025-11-21
> >
> > Thanks for the authors' response. I appreciate the additional experiments on the Villin protein, which further support AdaPath’s ability to generate physically plausible transition pathways.
> >
> > Regarding the generalization issue, I accept the authors' justification that the short training time makes it feasible to train a separate model for each system. That said, I would still like to see concrete statistics on the actual training time, particularly how it scales across systems of different sizes.
> >
> > I have also reviewed the revised manuscript, and the readability of the Methods section has indeed improved. However, I believe the organization of the Experiments section would also benefit from further refinement. Although alanine-dipeptide is a well-studied classical benchmark, it is a relatively simple peptide system. Table 1 may therefore occupy a disproportionate amount of space, whereas the much more challenging BPTI system receives comparatively limited discussion. This imbalance may hinder a proper assessment of the model's capability.
> >
> > Overall, I find AdaPath to be an original method with potential for real-world applications, while the manuscript's structural organization still has noticeable shortcomings. I will therefore maintain my score.

---

> > > ### Author Response · Authors · 2025-11-27
> > >
> > > Thank you for maintaining your positive evaluation and for the constructive feedback throughout the review process!
> > >
> > > We take your point about the organization of the experiment section. We'll review the presentation to identify ways to improve readability and better highlight the BPTI results. That said, we believe the detailed alanine dipeptide benchmark is necessary, given the strong emphasis multiple reviewers placed on a comprehensive comparison with existing methods.
> > >
> > > We thoroughly appreciate your engagement with the review process and our work!

---

### Official Review · Reviewer_scTv · 2025-10-29

**Soundness:** 2
**Presentation:** 3
**Contribution:** 2
**Rating:** 4
**Confidence:** 4

**Summary:**

The paper reformulates Minimum Energy Path (MEP) discovery as  a neural optimization problem. The transition pathways are parametrized with Implicit Neural Representations (INR) that are trained using a differentiable force-field and the proposed loss function. The loss function is derived from the transition path log probability and is essentially the batch-averaged potential energy along the transition pathway. A new INR architecture called AdaPath is proposed. The key feature of AdaPath is incorporating the progress coordinate $s$ via adaptive layer normalization and gating. Experiments are performed on two proteins: AIB9 and BPTI with explicit water molecules. For both proteins, the recovered transition pathways seem to be closely aligned with the results of long MD simulations.

**Strengths:**

The authors propose a scalable and efficient approach for MEP discovery. The experimental results demonstrate that trained INRs for a 58-residue protein with explicit water closely resemble states from long MD simulation performed in [1]. The training of INR requires $2 \times 10^{5}$ force field evaluations compared to $4 \times 10^{11}$ force field evaluations in MD simulation.

[1] Shaw, D. E., Maragakis, P., Lindorff-Larsen, K., Piana, S., Dror, R. O., Eastwood, M. P., ... & Wriggers, W. (2010). Atomic-level characterization of the structural dynamics of proteins. Science, 330(6002), 341-346.

**Weaknesses:**

- The loss function derived by authors does not seem to be well motivated or enforce continuity of the resulting transition pathway. The function $L = \frac{1}{B}\sum_{i=1}^B U(\phi_{\theta}(s_j))$ can be alternatively derived by assuming the independence of $x_{i+1}$ from $x_i$ (see Equations 3, 4):
$$
log P(x_0, \dots, x_N) = \{ \text{assume } x_0, \dots, x_N \text{ are independent}, x_i \sim \text{Boltzman}
\} = \sum_{i=1}^N log P(x_i) \propto \sum_{i=1}^N U(x_i).
$$
 When the transition pathway is reparametrized with $\phi_{\theta}$, we get the exact loss from AdaPath paper. However, the assumption of independence obviously does not hold for states in MEP as those must come form a continuous trajectory in the conformation space.
- The authors state a new AdaPath architecture and the new loss function as main contributions of the paper. Upon closer inspection, the derived loss seems to be a simplification of the loss functions introduced in [1]. However, the paper lacks experiments that demonstrate the benefits of the proposed loss function compared to loss function from [1]. Moreover, it would be great to see how the MLP architecture performs on AIB9 and BPTI with loss from [1].
- The proposed method’s efficiency is compared to the efficiency of classical methods only in terms of the number of force field evaluations. This does not take into consideration the computational resources required for the training of the neural network itself. Thus, it would be logical to compare those methods in terms of GPU-hours or wall-time.
- In Section D, the maximum energy for NEB is lower than the average energy, which seems to be impossible. This raises questions about the validity of baseline method results.

[1] Ramakrishnan, K., Schaaf, L. L., Lin, C., Wang, G., & Torr, P. (2025). Implicit Neural Representations for Chemical Reaction Paths. arXiv preprint arXiv:2502.15843.

**Questions:**

- The authors state that the validity of the proposed loss function approximation depends critically on maintaining sufficiently small displacement vectors. How is this ensured in practice?
- By removing the higher order gradient terms from the task defined be the loss function becomes ill-posed as it allows trivial solutions without continuity (see Section 2.1). To enforce continuity the authors have to carefully tune the optimizer and use gradient clipping. Both the Muon optimizer and the gradient clipping help maintain lower Lipshitz constant. I wonder if the method will become more robust to those hyperparameters if the loss from [1] is used?

[1] Ramakrishnan, K., Schaaf, L. L., Lin, C., Wang, G., & Torr, P. (2025). Implicit Neural Representations for Chemical Reaction Paths. arXiv preprint arXiv:2502.15843.

---

> ### Author Response · Authors · 2025-11-20
>
> We thank the reviewer for the positive overall assessment and for the insightful comments regarding the comparison with the INR-based formulation.
>
> Motivation and validity of the energy-based loss
>
> We rewrote the methods section to clarify the derivation and to better place it in context by first explaining classical approaches. To clarify the form of our loss function, which is indeed surprising when abstracted away from the parametrization. Across essentially all minimum-energy path (MEP) formulations, one optimizes an objective of the form:
>
> $ L(\varphi) = \sum_s U(\varphi(s)) + \lambda \, C(\varphi) $
>
> where the continuity term $C(\varphi)$ enforces smoothness or uniform spacing.
>
> Examples include:
>
> Chain-of-states (CoS), which uses a spring-type continuity penalty:
>
> $ C_{CoS}(\varphi) = \sum_k \| \varphi_{k+1} - \varphi_k \|^2 $
>
> String Method, which applies arc-length reparameterization:
>
> $ s_i = \frac{ \sum_{j=0}^{i-1} \| \varphi_{j+1} - \varphi_j \| }{ \sum_{j=0}^{N-1} \| \varphi_{j+1} - \varphi_j \| } $
>
> INR-based loss from Ramakrishnan et al. [1] subtracts the gradient in the direction along the MEP from the loss:
>
> $ C_{\text{INR}}(\theta) = - \frac{1}{B} \sum_{j=1}^B \mathrm{Stop} \big( \nabla U(\varphi_\theta(s_j))_{\parallel} \big)^{T}\varphi\_\theta(s_j) $
>
> In contrast, our method uses only the energy term:
>
> $ L(\theta) = \frac{1}{B} \sum_{j=1}^B U(\varphi_\theta(s_j))  $
>
> This is valid only if the path remains continuous, because the derivation uses the first-order Taylor approximation:
>
> $ U(\varphi_{k+1}) - U(\varphi_k) \approx \| \nabla U(\varphi_k) \| \, \| \varphi_{k+1} - \varphi_k \| \cos(\theta_k). $
>
> We now emphasize this requirement more explicitly. To validate it in practice, we added a path-velocity analysis. Across all systems, the ratio of maximum to mean velocity along the path matches that of classical approaches, indicating that AdaPath does not stretch or teleport, without explicit continuity constraints.
>
> Need for a direct comparison with the INR loss
>
> Following the reviewer’s suggestion, we implemented the INR loss [1] within AdaPath and performed ablation studies:
>
> - Alanine dipeptide: similar barriers, but faster convergence with our loss.
> - AIB9: INR loss shows high variance; our loss is more stable.
> - BPTI: our loss achieves substantially better barriers with fewer evaluations.
>
> Thus, it appears that the explicit subtraction of energy along the path directions helps in small systems, but the more complex loss mechanism seems to hurt in larger systems. Furthermore, evaluating the loss slows training by about 2 per batch.
>
> GPU-hours and wall-clock comparisons
>
> We added wall-clock timings:
>
> - AdaPath has a higher per-step cost but reaches high-quality paths in competitive or better total time on larger systems.
> - Classical methods take cheaper steps but converge to much higher-energy minima.
>
> Clarification of the NEB baseline
>
> The NEB mean energies appear high because we reparameterize all paths to have uniform arc-length spacing, making evaluations path-stretching-agnostic. NEB often produces jagged paths, so interpolation during arc-length respacing creates intermediate configurations that lie in high-energy regions, raising the arc-length–adjusted mean energy above the discrete maximum. We now explain this more clearly in the appendix.
>
> Summary of broader revisions
>
> We also added:
>
> - Committor analyses and path-velocity diagnostics,
> - A third biological system (Villin HP35 [2]) with extensive MD reference data.
>
> We again thank the reviewer for the valuable feedback. Given these extensive changes, we respectfully ask that you reconsider your score and raise it above the acceptance threshold.
>
> References
> [1] Ramakrishnan et al. (2025).
> [2] Piana et al. (2012), PNAS.

---

> > ### Comment · Reviewer_scTv · 2025-11-26
> >
> > Thank you for your detailed response and the additional experimental results. However, the questions raised in my review regarding (1) maintaining sufficiently small displacement vectors, and (2) enforcing the continuity of the path through hyperparameter selection, have not been directly addressed.
> >
> > Importantly, the first question relates to my main concern about the proposed loss function. In your response, you emphasized that “the approximation is valid only if the path remains continuous,” yet no explanation was provided as to why the path should remain continuous when the loss function does not include any term that explicitly enforces this property.

---

> > > ### Author Response · Authors · 2025-11-27
> > >
> > > Thank you for the follow-up. We address both questions directly.
> > >
> > > ### (1) How are sufficiently small displacement vectors maintained?
> > >
> > > The loss does not explicitly enforce continuity. It arises from and is validated by (i) the path parametrization, (ii) the AdaPath architecture, (iii) empirical diagnostics, and (iv) a theoretical argument about representability in high dimensions.
> > >
> > > 1. **Path parametrization (Eq. (21)).**
> > >    We use $x(s) = (1-s)x_A + s x_B + s(1-s)\\varphi\_\\theta(s)$, which forces the correction to vanish at the endpoints and ties interior updates to the linear interpolation. Given that the path starts as a linear interpolation and only modulates it, it is biased toward a continuous solution.
> > >
> > > 2. **Architecture (Section 2.2.1).**
> > >    AdaPath uses gated residual blocks that apply small corrections to shared atom embeddings as $s$ varies. Because the network predicts residual updates rather than absolute coordinates, the representation is biased to evolve continuously. This residual formulation and the parameter sharing are chosen based on gradient flow and efficient representation/scaling considerations and are thus not an imposition on the method.
> > >
> > > 3. **Path-velocity diagnostics (Appendix B.7).**
> > >    The max/mean velocity ratios are: alanine dipeptide $1.4 \\pm 0.6$, AIB9 $1.38 \\pm 0.20$, BPTI $1.7 \\pm 0.3$. Teleportation would give ratios $\gg 2$. This diagnostic shows that the first-order Taylor approximation used in the derivation remains valid during training.
> > >
> > > 4. **Theoretical argument (Petersen & Voigtlaender, 2018).**
> > >    There is also a representational argument: in high-dimensional spaces, standard neural networks have extreme difficulty approximating discontinuous functions unless the architecture is explicitly designed for this. As dimensionality grows, the parameter volume required to represent a discontinuity grows exponentially. This theoretical effect becomes stronger for larger molecular systems (Villin, BPTI), consistent with our empirical observation that discontinuities do not appear even without the Muon optimizer.
> > >
> > > Taken together, Eq. (21), the architecture in Section 2.2.1, the velocity statistics in Appendix B.7, and the representational argument above explain why the paths remain continuous in practice.
> > >
> > > ### (2) Would the INR loss be more robust?
> > >
> > > We implemented the INR loss (Ramakrishnan et al., 2025). Results appear in Appendix D.5 (Tables 20–22).
> > >
> > > 1. **Reduced robustness on larger systems.**
> > >    On AIB9, the INR loss exhibits much higher variance and consistently worse minima: INR Max Energy $743 \\pm 830$ kJ/mol, INR Mean $398 \\pm 206$ kJ/mol; AdaPath Max Energy $-496 \\pm 179$ kJ/mol, AdaPath Mean $-767 \\pm 24$ kJ/mol.
> > >    On BPTI, this gap widens substantially: INR MinMax $-3.2 \\times 10^4$ kJ/mol versus AdaPath MinMax $-1.4 \\times 10^5$ kJ/mol.
> > >    These results indicate that as dimensionality increases, the multi-term INR objective becomes harder to optimize to low-energy paths.
> > >
> > > 2. **More hyperparameters vs. standard regularization.**
> > >    The INR loss introduces additional weights $\\lambda\_s$ and $\\lambda\_{\\mathrm{cl}}$ for smoothness and climbing image loss terms.
> > >
> > > 3. **Computational overhead.**
> > >    INR requires tangent projections and path derivatives, thus the per-step cost increases by about 2.2×, and total evaluations rise from $6.4 \\times 10^4$ to $\\sim 1.1 \\times 10^6$.
> > >
> > > ### Summary
> > >
> > > - Continuity emerges from the parametrization, architecture, and the inherent representational bias of neural networks in high-dimensional settings. Gradient clipping helps stability due to the steep energy landscape, but it is standard practice in DL model training. The choice of the muon optimizer is only necessary on small systems, but given its recent rise as an alternative to Adam, this does not pose an imposition.
> > > - The absence of overstretching is supported empirically by ablations (Appendix D.2) and velocity statistics (Appendix B.7).
> > > - The INR loss (Appendix D.5) shows higher variance, worse performance on large systems, and higher computational cost. Most importantly, though, it introduces more hyperparameters, rather than removing them.
> > >
> > > We hope that these points, together, address the questions, and thank the reviewer for their engagement.

---

### Official Review · Reviewer_LogX · 2025-11-01

**Soundness:** 2
**Presentation:** 1
**Contribution:** 2
**Rating:** 4
**Confidence:** 3

**Summary:**

This paper proposes AdaPath, a fast and scalable architecture for finding minimum energy paths. The framework is optimized with the loss function from the path’s likelihood with the Onsager-Machlup action, and validated for two proteins.

**Strengths:**

1. Experiments on large biomolecules (Significance)

While prior works mainly have tackled the problem on smaller biomolecules with lengths of 10~20 residues, the paper additionally tackles BPTI with 58 residues.

**Weaknesses:**

1. Lack of comparison with ML methods

The paper lacks detailed comparison with prior works. Though authors have cited some prior works, I missed points on why some baselines are missing. They should denote clearly if the prior works fail to scale to larger systems, or impossible to compare for a reason

- A similar work solve transition path sampling target with Onsager_machlup functional [1]. The authors should states the difference between them.

- Though I understand that prior works mostly generate ensembles of transition paths rather than a minimum energy path, the authors should still denote the results [2-5].

- While the Doobs Seq2Seq [4] and Doobs’ Lagrangian [5] is different, the authors include only one of them across table 1 and table 2, for a different system. Both should be stated.

2. Experiments - physical validation on transition states

While the computed MEP from AdaPath does not get off the low energy region dramatically, additional validation on the generated path would strengthen the paper a lot. For example, measuring the dynamics content, energy along the MEP would improve validity on the MEP. If a committor function analysis is possible (there might not be one for every molecular system, I understand even if this is missing), doing them on intermediate states would be great.

3. Presentation (clarity)

I think the paper could improve presentation in some aspects. The most important results, quantitative results in table 1 to table 3 are all in the appendix. Making some space in the main paper an putting them would be good. section 2.2 on related works could summarized briefly to discuss the difference of AdaPath to prior works, and details to the appendix.

4. Lack of evidence for the claim fast

The authors claim a fast and scalable neural optimization problem. However, I do not find any results showing that the proposed method is faster than prior works. Additionally, what is the exact time component the authors claim for fast?

[1] Action-Minimization Meets Generative Modeling: Efficient Transition Path Sampling with the Onsager-Machlup Functional, ICML 2025

[2] Stochastic Optimal Control for Collective Variable Free Sampling of Molecular Transition Paths, NeurIPS 2023

[3] Transition Path Sampling with Improved Off-Policy Training of Diffusion Path Samplers, ICLR 2024

[4] Scaling Deep Learning Solutions for Transition Path Sampling, preprint

[5] Doob’s Lagrangian: A Sample-Efficient Variational Approach to Transition Path Sampling, NeurIPS 2024

Minor

- Seems like section 4, 5 should be section 3.2, 3.3 regarding that 3.1 is for a system type?
- line 241 “shared transformer-like MLP blocks with ada”, this means that a single MLP is used for every coordinates?

**Questions:**

1. Scalability claim

One of the main contribution the authors are claiming is scalability. Is this scalability only for the system size? If so, there are no problems, but if something different, I do not see any results to support it

2. Architecture details

Could the authors explain more about the architecture? I understood what the task is and what the input & output of the model, but how is the AdaLN blocks incorporated? The original adaLN blocks were used diffusion transformers [1], replacing the standard norm layer with scale and shift parameters. Why is the up and down MLP needed?

---

> ### Author Response · Authors · 2025-11-20
>
> We thank the reviewer for the constructive feedback. Below, we aim to address points.
>
> 1. Need for clearer comparison with ML-based approaches
>
> We expanded the comparison to prior ML-based methods and added explanations of why we benchmarked each method in the benchmark section of the appendix and the experiments section. :
>
> - The paper now includes additional results for the INR-style loss of Ramakrishnan et al. [1] when used with AdaPath.
> - A new appendix section clarifies which ML baselines we include, why certain approaches, such as diffusion-model–based force-field surrogates [2], cannot be compared fairly, and how differences in problem formulation limit direct comparisons.
> - We additionally discuss methods such as Lee et al. [3]. Regrettably, no public implementation of this relatively complex architecture exists, which makes a faithful benchmark infeasible beyond the systems Alanine Dipeptide that they themselves benchmarked and reported results for.
>
> 2. Request for additional physical validation
>
> We added several layers of physical validation:
>
> - For alanine dipeptide and AIB9, we performed committor probability analyses starting from configurations along the learned MEP. This verifies that the learned path is physically meaningful enough to start stable MD simulations from, and that the progress coordinate maps smoothly to the committor, abteil in a nonlinear way, as expected since they are related but different quantities.
> - We added energy profiles for all smaller systems where they were previously missing (alanine dipeptide and AIB9).
> - We added a new experiment on Villin HP35, a protein with extensive MD reference data. The recovered MEP aligns well with known features of the free-energy landscape, further supporting the physical plausibility of our approach.
>
> 3. Placement and visibility of key results
>
> We improved the presentation of core results:
>
> - The most important benchmark tables have been moved from the appendix to the main paper.
> - We streamlined the related-work discussion and moved extended comparisons to the appendix.
>
> 4. Clarifying the “fast” and “scalable” claims
>
> We clarified our terminology:
>
> - Scalable refers specifically to scaling with respect to system size (atom count).
> - Fast refers to producing high-quality transition paths orders of magnitude faster than MD simulations that directly observe such transitions. We initially did not want to make claims about speed relative to classical baselines. However, studying wallclock times is extremely informative, and we thank the reviewer for pointing out that we should include them. We thus added explicit wall-clock timings for all methods on a single NVIDIA A6000 GPU:
>
> - AdaPath incurs a higher per-step cost due to backpropagation, but nevertheless reaches high-quality paths in competitive or shorter total time on large systems.
> - On smaller systems, classical methods may take cheaper steps but still converge to significantly higher-energy paths.
> - On large systems such as BPTI, AdaPath finds state-of-the-art paths in minutes, whereas classical methods require substantially more evaluations and still fail to reach comparable quality.
>
> 5. Improved clarity of the architecture description
>
> We fully reworked the architecture section:
>
> - The forward computation is now explained step by step.
> - We detail how adaptive layer normalization (AdaLN) enables progress-conditioned readout of atom embeddings.
> - We clarify the roles of AdaLN, embedding structure, and coordinate-dependent modulation.
>
> This revision substantially improves readability and makes the architectural contribution clearer; the section is now greatly expanded in detail.
>
> 6. Summary of broader revisions
>
> In addition to the points above, we also:
>
> - Added detailed path-velocity diagnostics,
> - Moved key benchmark tables to the main paper and clarified success rates across training seeds.
>
> We thank the reviewer again for the helpful suggestions. Given the extensive improvements addressing ML comparisons, physical validation, and architectural clarity, we respectfully ask that you consider raising the score above the acceptance threshold.
>
> References
> [1] Ramakrishnan et al. (2025).
> [2] Raja et al. (2025).
> [3] Lee et al. (2025), Scaling Deep Learning Solutions for Transition Path Sampling.

---

> > ### Comment · Reviewer_LogX · 2025-11-28
> >
> > I thank the authors for the additional experiments.
> >
> > - [W1] additional results from ML methods, and explaining why fair comparison is impossible for some. However, some important works seems to be still missing.
> > - [W2] additional committor probability analyses, energy profile figures of the transition pathway
> > - [W3, Q2] improved results and architecture description presentation
> > - [W4, Q1] Clarification on scalable and fast, with wall-clock time comparison
> >
> > I believe that the additional results would also be organized and presented in future. I will raise my score accordingly since the edit button has vanished, as soon as the technical issues is resolved.
> >
> > I still have some additional questions and concerns extend from the original weakness and questions.
> >
> > [W1] Related works
> >
> > - Just checking, Doob’s Cartesian and Doob’s Internal in Table 1 refers to the variants in Doob’s Lagrangian, right?
> > - Among [1-5] papers of my initial review, while Yan et al. and [2] has been excluded since they are learning external force methods, there are no mention of [3]. Since [3] is also a biasing force learning method, it should be added to this part, or is there another reason for exclusion?
> >
> > [W3,Q2] Architecture details regarding rotation
> >
> > - I understood that the 3d coordinates are decided by the $s$ interpolation on the endpoints and the output of the model. How is the rotation handled? For a rotated endpoint, does the network be trained again? I may have missed the details, could it simple be resolved by applying a Kabsch algorithm to a reference frame?

---

> ### Author Response · Authors · 2025-11-28
>
> We thank the reviewer for acknowledging the improvements and for their willingness to raise the score. We address the remaining questions below.
>
> Related works clarifications [W1]:
>
> Doob's variants: Yes, "Doob's Cartesian" and "Doob's Internal" in Table 1 refer to the coordinate-space variants from the Doob's Lagrangian paper [1]. Specifically, these determine whether the interpolation and model corrections operate in Cartesian or internal (dihedral angle) coordinate space. The internal coordinate formulation provides an inductive bias for single-molecule systems, as it naturally respects bond lengths and angles. However, this approach does not generalize to multi-component systems such as explicitly solvated proteins, where internal coordinates cannot be defined across solvent molecules. This is why we chose to work entirely in Cartesian space throughout our method. We will clarify the different Doob's variants in the Benchmark section (Appendix B).
>
> Diffusion Path Samplers: We thank the reviewer for pointing us toward this work [2]. It is indeed exciting and relevant to the broader landscape of transition path methods. We will add a discussion of this paper to our benchmark scope section (Appendix B.1). However, since this method also learns a drift/biasing force rather than explicitly parameterizing transition paths, direct quantitative comparison faces the same challenges as with Holdijk et al. [3] and Yan et al. [4]: the learned quantities (control policies vs. path geometries) and evaluation frameworks differ fundamentally.
>
> Rotation handling [W3/Q2]:
>
> The network does not need to be retrained for rotated endpoints. We handle rotational invariance when preparing the endpoints and while training as follows:
>
> Endpoint alignment: As described in Appendix F (AIB9) and Appendix G (BPTI), we apply Kabsch alignment to optimally superimpose the end configuration onto the start configuration before training. This removes translational and rotational degrees of freedom from the endpoint boundary conditions and is performed once as a preprocessing step.
>
> Intermediate configurations: The linear interpolation between endpoints maintains a fixed orientation throughout the path. While the neural network outputs corrections to this interpolation, these corrections could, in principle, introduce rotations. To prevent the network from having to learn to compensate for such rotations when predicting atomic displacements, we rotationally align the model output back to the interpolation baseline. This ensures the network focuses purely on learning the chemically relevant conformational changes rather than rigid-body motions.
>
> Extended committor analysis [W2]:
>
> Following up on the reviewer's original suggestion for committor analysis, which we found very valuable, we extended our empirical trajectory-shooting results using an analytical approach. We use a closed-form expression for the committor based on the one-dimensional energy profile along the MEP, using the steady-state backward Fokker-Planck equation under overdamped dynamics. This analytical committor agrees well with our empirical results (Figure 10, Appendix G.1.4) and provides a computationally efficient alternative to extensive shooting simulations for identifying transition states. The reviewer might find this extension interesting, given that their suggestion motivated this direction.
>
> We thank the reviewer again for the constructive engagement throughout the review process.
>
> References
>
> [1] Du et al. Doob's Lagrangian: A Sample-Efficient Variational Approach to Transition Path Sampling, NeurIPS 2024.
>
> [2] Seong et al. Transition Path Sampling with Improved Off-Policy Training of Diffusion Path Samplers, ICLR 2025.
>
> [3] Holdijk et al. Stochastic Optimal Control for Collective Variable Free Sampling of Molecular Transition Paths, NeurIPS 2023.
>
> [4] Yan et al. Learning nonequilibrium control forces to characterize dynamical phase transitions, Physical Review E.

---

### Official Review · Reviewer_jWgn · 2025-11-04

**Soundness:** 1
**Presentation:** 1
**Contribution:** 2
**Rating:** 2
**Confidence:** 5

**Summary:**

In this work, the authors present a neural representation-based approach to find the Minimum-Energy-Path (MEP) between two molecular conformations, an important problem in computational chemistry. For this purpose, the authors derive a loss function based on a relaxation of the Onsager–Machlup action functional. In short, the derived loss function reduces to a sum over the potential energy along the path at discretized intervals.

To learn the neural representation, the authors present their key architecture, “AdaPath,” whose primary contribution is that it uses shared atom embeddings. The authors claim that this shared embedding is one of the main reasons for the success of their introduced method. The output of AdaPath is an offset to the linear interpolation connecting the endpoints. In the section introducing AdaPath, the authors discuss an important possible failure point of the method in the form of path stretching and discuss how this should be resolved by the implicit smoothness of neural architectures.

After an in-depth discussion of other approaches to MEP, the authors present their experimental evaluation of their method using the AIB9 and BPTI systems. Compared to the general literature on MEP, these systems are of considerable size. The presented results show that the found MEPs closely align with the FEP landscape in the case of AIB9 and follow the same intermediate states as found in other large-scale studies for BPTI.

**Strengths:**

With their specific focus on MEP discovery, the authors target an interesting and very important problem in computational biochemistry. While recently there has been significant focus on problems such as equilibrium and transition path sampling for molecular systems, MEP is still somewhat overlooked despite its importance.

The main strengths of the paper are in the scalability of the presented method, as exemplified by the experiments conducted. As stated in the paper, the size of the systems considered is significantly larger than is normally considered for MEP approaches or even related problems such as Transition Path Sampling.

For this reason alone, the paper already makes a good case for publication at ICLR. However, as highlighted below, there remain a few issues and questions that need to be clarified and/or resolved before I would feel fully comfortable voting for acceptance.

**Weaknesses:**

As discussed above, while I believe the paper has significant contribution and novelty, it currently suffers from two major issues that make me reluctant to vote for acceptance. First, the paper is not sufficiently structured and is hard to follow at times; and second, while the experimental results are impressive, they currently do not sufficiently validate that the main failure mode of path stretching is fully resolved by the smoothness of neural architectures.

Regarding the first point, as it stands, the paper would significantly benefit from more structure and formalism. While it currently contains a long derivation of the proposed loss function in Section 2.1, the lack of clear subheadings and paragraphs makes it hard to follow the flow. Adding some clear definition and theorem environments to this section would also help. The same holds for the section on AdaPath, to highlight when different components/limitations are discussed. Moving the formal overview of established approaches before the derivation of the loss function would also help, as would a short formal definition of the MEP problem at this early stage of the paper. Lastly, the introduction would benefit a lot from some additional clear subheadings, as it is currently very long.

Regarding the path-stretching failure mode, as stated, I am currently insufficiently convinced that this is fully resolved by the implicit smoothness of neural architectures. While the presented experimental results are impressive in terms of scaling, they make it hard to study such core failure modes. To resolve these issues, I believe moving the experimental results based on Alanine Dipeptide to the main body of the paper would greatly help, if accompanied by a further in-depth study of the found transition path. For example, it would be useful to see how the velocity along the path for AdaPath compares to that of traditional methods. If the presented method does indeed not suffer from the path-stretching failure mode, the maximum velocities should be roughly the same.

I believe both of these points, the structure of the paper and the experimental results, are necessary for me to be able to increase my score.

In addition to this, I have a few additional comments below, but I do not deem these as hard requirements for acceptance and/or think they should be relatively easy to address:
- The last sentence of the abstract has a very large claim that is, in my opinion, not sufficiently justified.
- Lines 35–38 are overly simplified and generally do not read well.
- The switch to a discussion about ML-based MEP methods in line 104 is abrupt and generally hard to follow.
- Figure 1 is placed a few pages earlier than it is first referenced. While I understand the placement of the figure here, it would be better to at least have a short reference to the figure on the same page. Aside from this, when referenced, the writing only says “1,” not “Fig. 1” or “Figure 1.”
- It should be made clearer in the experimental section that the results of different spline MEPs, as visualized in Figure 2, are due to the training of different models and not due to variation within the model from different seeds.
- It is unclear if all training runs result in successfully trained models; it would be good to get this clarified in the paper.
- When discussing the computational efficiency of the method (lines 417–422), the training time of each model should be included.

**Questions:**

See weaknesses

---

> ### Author Response · Authors · 2025-11-20
>
> We thank the reviewer for the thoughtful and detailed feedback, and for noting that our work “already makes a good case for publication at ICLR.” Below, we hope to address all major points.
>
> 1. Need for clearer structure and more formalism
>
> We fully agree and have substantially reorganized the manuscript:
> - The overview of classical MEP approaches (string, NEB, chain-of-states) now appears before our derivation, providing context and highlighting how our formulation differs.
> - The derivation is rewritten with clear subsections containing explicit definitions, assumptions, and short propositions, replacing the previous long block of text.
> - The AdaPath architecture section is reorganized and greatly expanded into well-separated subsections (atom embeddings, progress conditioning, AdaLN blocks, endpoint constraints, parameter efficiency).
> - The Introduction has been rewritten with clearer paragraphs and content coupling and improved transitions, making the narrative easier to follow.
> - As suggested, we moved the alanine dipeptide experiment into the main paper.
> These changes directly address the concerns about readability and formal structure.
>
> 2. Concern about path stretching and missing validation
>
> We implemented the suggested diagnostic and added a complete path-velocity analysis.
> We compute the magnitude of the spatial derivative along the path and report the ratio of the maximum to the mean velocity.
> Large ratios would indicate stretching; values near 1–2 indicate well-behaved paths.
>
> Across all systems, AdaPath matches classical baselines:
> - Alanine dipeptide: AdaPath ≈ 1.4 ± 0.6 (String ≈ 1.0, NEB ≈ 1.42)
> - AIB9: AdaPath ≈ 1.38 ± 0.20 (COS ≈ 1.05, NEB ≈ 1.08)
> - BPTI: AdaPath ≈ 1.7 ± 0.3 (NEB ≈ 1.7)
>
> This confirms that AdaPath does not exhibit pathological stretching and behaves comparably to classical methods, while still achieving better energy barriers. The main text summarizes this, and full plots are provided in the appendix.
>
> 3. Further points on clarity, figure placement, and training stability
>
> - The final sentence of the abstract was softened for accuracy. We would also like to clarify that the D. E. Shaw simulation we use as a reference only yielded a single observed transition, not an equilibrium distribution, so the comparison is more appropriate than comparing equilibrium MD to a MEP method.
> - The Introduction has been reorganized so that classical MEP methods, classical sampling approaches, and ML-based methods are each discussed in distinct, well-separated paragraphs.
> - Figure references now use consistent “Figure X” notation, and Figure 1 is placed near its first mention.
> - The caption and text for Figure 2 now explicitly state that the shown spline paths are from different model training runs with distinct initial conditions and seeds.
> - Convergence across seeds is rather reliable due to log-scaled energy and the updated architecture, as reflected in the benchmarks’ low average energies (not just minima).
> - We added wall-clock timings on a single A6000 GPU, enabling direct comparison in GPU minutes rather than relying solely on force-field evaluations. AdaPath is broadly comparable in speed to the fastest classical methods on both small and large systems (with AIB9 as the main exception) and is substantially more accurate.
>
> 4. Summary of broader revisions
>
> In addition to addressing the above points, we also:
> - Added committor analyses and energy profiles to validate the physical relevance of the paths.
> - Implemented and compared against the INR loss of Ramakrishnan et al. [1], showing that our simpler loss is more stable and more effective on larger systems.
> - Added the Villin HP35 fast-folding protein to demonstrate applicability to folding transitions on a well-validated system, with existing reference MD data.
> - Moved other key benchmark tables to the main paper.
> These revisions required substantial additional experiments and analysis, and we hope they significantly strengthened the clarity, rigor, and breadth of the work.
>
> We thank the reviewer again, especially for the suggestion to analyze path velocity and for pointing out the clarity issues, which substantially improved the paper. In light of the extensive changes we made to address your primary concerns, we respectfully ask that you reconsider your score.
>
> References
> [1] Ramakrishnan, K., Schaaf, L. L., Lin, C., Wang, G., & Torr, P. (2025). *Implicit Neural Representations for Chemical Reaction Paths.* arXiv:2502.15843.

---

### Author Response · Authors · 2025-12-02

**Summary for Area Chair**

We appreciate you taking on this submission on short notice, given the circumstances. We have condensed the key points from the review process below.

We present a neural method that discovers protein transition mechanisms in minutes on a single GPU, matching results from millisecond-scale MD simulations that required weeks on specialized hardware, the first to scale neural MEP discovery to explicitly solvated systems with 3,500+ atoms.

**Core Revisions Made During Rebuttal**

- Added a fourth protein system (Villin HP35 fast-folder) demonstrating applicability to protein folding transitions.
- Added committor probability analyses with both empirical trajectory shooting and a new analytical derivation for physical validation, as well as verifying the ability of the learned MEP to serve as a basis for further sampling.
- Added path-velocity analysis confirming our method does not exhibit pathological "teleportation" behavior, a core methodological concern.
- Implemented and benchmarked the INR loss [1] as a direct comparison, showing our simpler loss is more stable on large systems.
- Added wall-clock timing comparisons contextualizing the speed of our method compared to classical MEP approaches, showing how the method results in comparable optimization times to classical methods while yielding superior results, despite the overhead of the neural parametrization.
- Restructured the paper: clearer derivation with definitions/propositions, reorganized and expanded architecture section, moved key benchmark tables to the main text.
- Expanded discussion of ML baselines and scope of comparisons.

**Reviewer Status at Time of Incident**

- **Reviewer LogX (score 4)**: Main concerns were ML method comparisons, physical validation, and evidence for "fast" claims. We addressed all points with expanded baselines, committor analyses, and wall-clock timings. Explicitly stated "I will raise my score accordingly" but could not due to the bug [see their Nov 28 comment].
- **Reviewer jWgn (score 2)**: Main concerns were paper structure and path-stretching validation. Noted that "for this reason alone [scalability], the paper already makes a good case for publication at ICLR." Listed two specific requirements for raising their score: (1) improved structure/formalism, (2) path-velocity analysis. Both were addressed comprehensively.
- **Reviewer scTv (score 4)**: Main concerns were loss function motivation, comparison with INR loss, and wall-clock timing. All questions addressed, including full INR loss implementation and timing comparisons.
- **Reviewer 9Rgq (score 6)**: Main concerns were training robustness and the need for more biological examples. Described AdaPath as "an original method with potential for real-world applications." Maintained a positive score after we added Villin HP35 experiments.

All requested experiments and revisions were completed during the rebuttal period.

**References**

[1] Ramakrishnan et al. (2025), Implicit Neural Representations for Chemical Reaction Paths.

---

### Meta-Review · Area_Chair_cZy1 · 2026-01-05

**Summary:**

This paper discusses methods for searching for Minimum Energy Paths (MEPs) between molecular conformations. Unlike related works that involve probability-based path sampling, this article directly addresses the energy minimization problem for interpolated paths.
The approach combines ideas from two existing works: [1] which proposed a neural network-based continuous path parameterization, and [2] which proposed a transition path sampling method using machine learning force fields and direct discrete path parameterization. The authors perform path optimization by applying the path parameterization from [1] and the objective from [2] under a given classical differentiable molecular force field.

Strengths:
Beyond merely combining these methods, the authors' primary contribution lies in designing a new path parameterization using the AdaLN network architecture found in diffusion transformers, and simplifying the Onsager-Machlup action into an integral form within the Geometric action via relaxation. The authors claim these improvements enhance scalability and reduce the number of force field evaluations, a claim supported by experiments on large-scale molecular datasets.

Weaknesses:
(1) Unreasonable Relaxation of the Onsager-Machlup Action: From the perspective of the Onsager-Machlup action itself, the authors' relaxation of the loss is unreasonable. In processing the action, the authors replaced the penalty on the absolute difference of potential energy with a penalty on the absolute value of the potential energy. While the inequality underlying this relaxation may hold mathematically, this substitution is logically unsound for the problem at hand and is difficult to accept. Obviously, optimizing the absolute value of a function can optimize the difference, but this fundamentally alters the behavior of the function's values and may lead to the collapse of molecular conformations. Although this is a molecular dynamics paper, to illustrate this point intuitively, I refer to a computer graphics paper regarding the prevention of self-intersection in shell interpolation [3]. Figure 3 in that work demonstrates that directly penalizing the repulsion potential causes the surface to "explode" away.

(2) Necessity to Reformulate Optimization Objectives: Even if the authors' relaxation has some validity within molecular dynamics, please note that the authors are evidently applying a different loss function in practice that forces energy minimization rather than minimizing the absolute value of energy changes. These two optimization objectives are by no means equivalent. This could lead to an alternative continuous interpretation corresponding to a functional loss outside of the Onsager-Machlup action. The authors must treat this issue seriously, as the corresponding optimizations yield drastically different behaviors. Therefore, even if the relaxation might be justifiable, the authors must rigorously re-examine the resulting loss function and reformulate the theoretical section. Furthermore, Reviewer scTv pointed out potential contradictions in the physical assumptions derived from this loss, which the authors should take seriously.

In summary, although the paper demonstrates a certain degree of completeness, the theoretical defects in the main text cannot be ignored. The authors need to undertake major revisions for the paper in a future review cycle.

[1]: Kalyan Ramakrishnan, Lars L. Schaaf, Chen Lin, et al. "Implicit Neural Representations for Chemical Reaction Paths."
[2]: Sanjeev Raja, Martin Sipka, Michael Psenka, et al. "Action-Minimization Meets Generative Modeling: Efficient Transition Path Sampling with the Onsager-Machlup Functional."

**Reviewer Concerns:**

The authors addressed some of the reviewers' concerns. Specifically:

(1) The authors restructured the theoretical framework of the article according to Reviewer jWgn's suggestions and provided evidence regarding path continuity as requested.

(2) Reviewer LogX primarily focused on the comparison of ML methods, which is supplemented by the authors during the rebuttal.

(3) The authors provided additional biological examples as requested by Reviewer 9Rgq.

However, several issues remain unresolved. When Reviewer scTv questioned the motivation behind the loss design, the authors attempted a derivation from a finite difference perspective. This relaxation is unreasonable from such a perspective. Reviewer scTv’s account of the physical meaning and the corresponding physical assumptions appears more rational; however, after Reviewer scTv pointed out contradictions in the derivation arising from these physical assumptions, the authors failed to provide a direct response.

**Reviewer Scores:**

Following the discussion, Reviewer LogX stated, "I will raise my score accordingly," but was unable to update the score due to policy constraints. Based on the authors' response to Reviewer LogX, his score is deemed to be 6. Since Reviewer jWgn’s concerns were partly addressed, his score could be increased to 4. Therefore, the final scores could be: Reviewer jWgn (4), Reviewer LogX (6), Reviewer scTv (4), Reviewer 9Rgq (6).

---

### Decision · Program_Chairs · 2026-01-26

Reject